# Simultaneous enumeration of cancer and immune cell types from bulk tumor gene expression data

Julien Racle[1,2], Kaat de Jonge[3], Petra Baumgaertner[3], Daniel E Speiser[3], David Gfeller[1,2]*

[1]Ludwig Centre for Cancer Research, Department of Fundamental Oncology, University of Lausanne, Epalinges, Switzerland; [2]Swiss Institute of Bioinformatics, Lausanne, Switzerland; [3]Department of Fundamental Oncology, Lausanne University Hospital (CHUV), Epalinges, Switzerland

**Abstract** Immune cells infiltrating tumors can have important impact on tumor progression and response to therapy. We present an efficient algorithm to simultaneously estimate the fraction of cancer and immune cell types from bulk tumor gene expression data. Our method integrates novel gene expression profiles from each major non-malignant cell type found in tumors, renormalization based on cell-type-specific mRNA content, and the ability to consider uncharacterized and possibly highly variable cell types. Feasibility is demonstrated by validation with flow cytometry, immunohistochemistry and single-cell RNA-Seq analyses of human melanoma and colorectal tumor specimens. Altogether, our work not only improves accuracy but also broadens the scope of absolute cell fraction predictions from tumor gene expression data, and provides a unique novel experimental benchmark for immunogenomics analyses in cancer research (http://epic.gfellerlab.org).

DOI: https://doi.org/10.7554/eLife.26476.001

*For correspondence: david.gfeller@unil.ch

Competing interests: The authors declare that no competing interests exist.

## Introduction

Tumors form complex microenvironments composed of various cell types such as cancer, immune, stromal and endothelial cells (*Hanahan and Weinberg, 2011*; *Joyce and Fearon, 2015*). Immune cells infiltrating the tumor microenvironment play a major role in shaping tumor progression, response to (immuno-)therapy and patient survival (*Fridman et al., 2012*). Today, gene expression analysis is widely used to characterize tumors at the molecular level. As a consequence, tumor gene expression profiles from tens of thousands of patients are available across all major tumor types in databases such as Gene Expression Omnibus (GEO [*Edgar et al., 2002*]) or The Cancer Genome Atlas (TCGA [*Hoadley et al., 2014*]). Unfortunately, flow cytometry or immunohistochemistry (IHC) measurements to quantify the number of both malignant and tumor-infiltrating immune cells are rarely performed for samples analyzed at the gene expression level. Therefore, to correctly interpret these data in particular from an immuno-oncology point of view (*Angelova et al., 2015*; *Gentles et al., 2015*; *Hackl et al., 2016*; *Li et al., 2016*; *Linsley et al., 2015*; *Rooney et al., 2015*; *Şenbabaoğlu et al., 2016*; *Zheng et al., 2017*), reliable and carefully validated bioinformatics tools are required to infer the fraction of cancer and immune cell types from bulk tumor gene expression data.

To this end, diverse bioinformatics methods have been developed. Some aim at estimating tumor purity based on copy number variation (*Carter et al., 2012*; *Li and Li, 2014*), or expression data (*Ahn et al., 2013*; *Clarke et al., 2010*; *Quon et al., 2013*; *Yoshihara et al., 2013*), but do not provide information about the different immune cell types. Others focus on predicting the relative

**eLife digest** Malignant tumors do not only contain cancer cells. Normal cells from the body also infiltrate tumors. These often include a variety of immune cells that can help detect and kill cancer cells. Many evidences suggest that the proportion of different immune cell types in a tumor can affect tumor growth and which treatments are effective.

Researchers often study tumors by measuring the expression of genes, i.e., which genes are active in tumors. However, the proportion of different cell types in the tumor is often not measured for tumors studied at the gene expression level.

Racle et al. have now demonstrated that a new computer-based tool can accurately detect all the main cell types in a tumor directly from the expression of genes in this tumor. The tool is called "Estimating the Proportion of Immune and Cancer cells" – or EPIC for short. It compares the level of expression of genes in a tumor with a library of the gene expression profiles from specific cell types that can be found in tumors and uses this information to predict how many of each type of cell are present. Experimental measurements of several human tumors confirmed that EPIC's predictions are accurate.

EPIC is freely available online. Since the active genes in tumors from many patients have already been documented together with clinical data, researchers could use EPIC to investigate whether the cell types in a tumor affect how harmful it is or how well a particular treatment works on it. In the future, this information could help to identify the best treatment for a particular patient and may reveal new genes that cause malignant tumors to develop and grow.

DOI: https://doi.org/10.7554/eLife.26476.002

proportions of cell types by fitting reference gene expression profiles from sorted cells (*Gong and Szustakowski, 2013*; *Li et al., 2016*; *Newman et al., 2015*; *Qiao et al., 2012*) or with the help of gene signatures (*Becht et al., 2016*; *Zhong et al., 2013*). These approaches have been recently applied to cancer genomics data to investigate the influence of immune infiltrates on survival or response to therapy (*Charoentong et al., 2017*; *Gentles et al., 2015*; *Şenbabaoğlu et al., 2016*) or predict potential targets for cancer immunotherapy (*Angelova et al., 2015*; *Li et al., 2016*). However, none of these methods provides quantitative information about both cancer and non-malignant cell type proportions directly from tumor gene expression profiles. In addition, reference gene expression profiles used in previous studies have been mainly obtained from circulating immune cells sorted from peripheral blood and were generally based on microarrays technology. Finally, several of these approaches have not been experimentally validated in solid tumors from human patients.

Here, we developed a robust approach to simultaneously Estimate the Proportion of Immune and Cancer cells (EPIC) from bulk tumor gene expression data. EPIC is based on a unique collection of RNA-Seq reference gene expression profiles from either circulating immune cells or tumor- infiltrating non-malignant cell types (i.e., immune, stromal and endothelial cells). To account for the high variability of cancer cells across patients and tissue of origin, we implemented in our algorithm the ability to consider uncharacterized, possibly highly variable, cell types. To validate our predictions in human solid tumors, we first analyzed melanoma samples with both flow cytometry and RNA-Seq. We then collected publicly available IHC and single-cell RNA-Seq data of colorectal and melanoma tumors. All three validation datasets showed that very accurate predictions of both cancer and non-malignant cell type proportions could be obtained even in the absence of *a priori* information about cancer cells.

## Results

### Reference gene expression profiles from circulating and tumor-infiltrating cells

EPIC incorporates reference gene expression profiles from each major immune and other non-malignant cell type to model bulk RNA-Seq data as a superposition of these reference profiles (*Figure 1A,B*). To tailor our predictions to recent gene expression studies, we first collected and curated RNA-Seq profiles of various human innate and adaptive circulating immune cell types

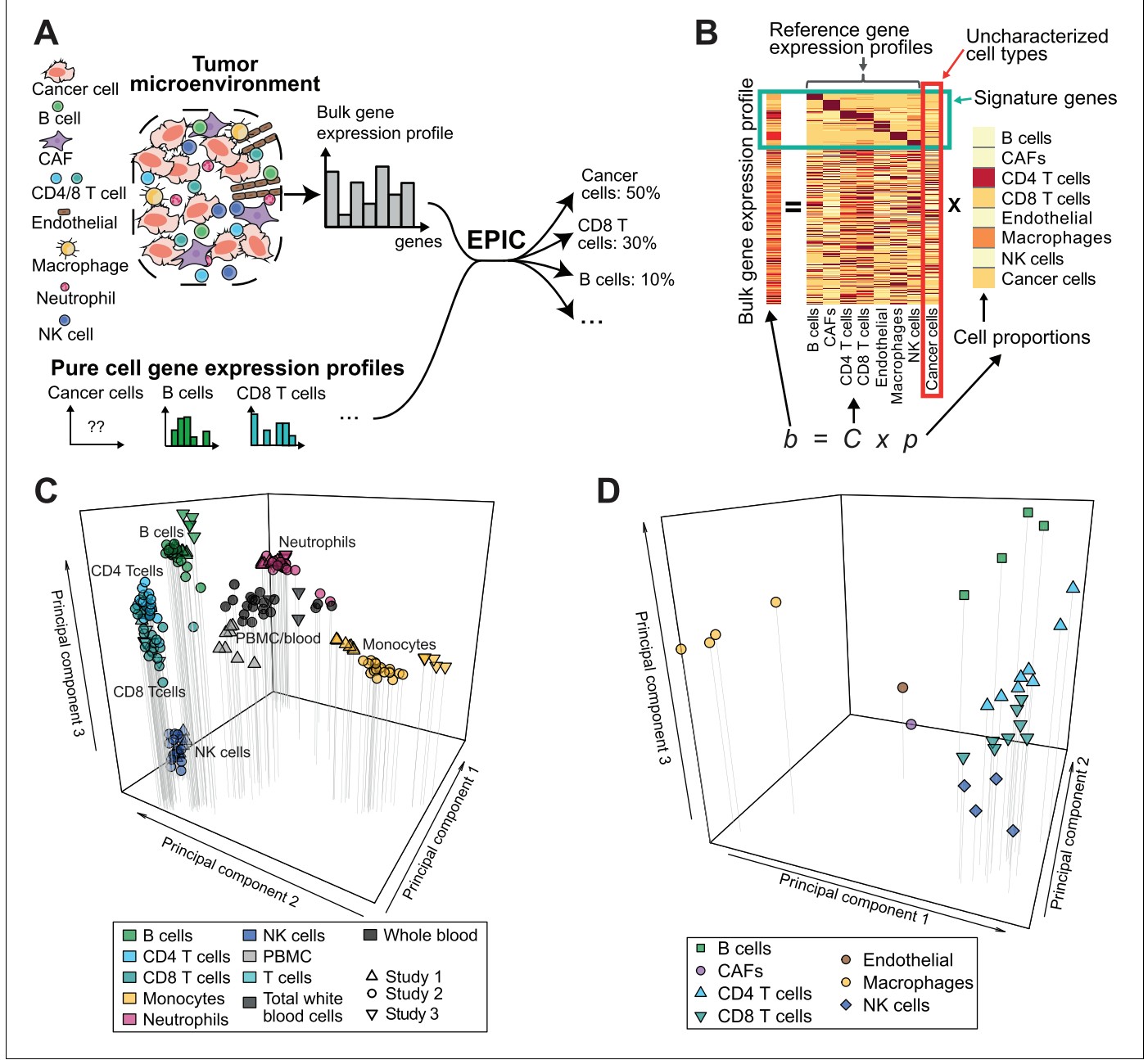

**Figure 1.** Estimating the proportion of immune and cancer cells. (**A**) Schematic description of our method. (**B**) Matrix formulation of our algorithm, including the uncharacterized cell types (red box) with no or very low expression of signature genes (green box). (**C**) Low dimensionality representation (PCA based on the 1000 most variable genes) of the samples used to build the reference gene expression profiles from circulating immune cells (study 1 [*Hoek et al., 2015*], study 2 [*Linsley et al., 2014*], study 3 [*Pabst et al., 2016*]). (**D**) Low dimensionality representation (PCA based on the 1000 most variable genes) of the tumor- infiltrating cell gene expression profiles from different patients. Each point corresponds to cell-type average per patient of the single-cell RNA-Seq data of *Tirosh et al. (2016)* (requiring at least 3 cells of a given cell type per patient). Only samples from primary tumors and non-lymphoid tissue metastases were considered. Projection of the original single-cell RNA-Seq data can be found in *Figure 1—figure supplement 1*.
DOI: https://doi.org/10.7554/eLife.26476.003

The following figure supplements are available for figure 1:

**Figure supplement 1.** Low dimensionality representation of the tumor-infiltrating cell samples.
DOI: https://doi.org/10.7554/eLife.26476.004
**Figure supplement 2.** Cell type mRNA content.
DOI: https://doi.org/10.7554/eLife.26476.005

(*Hoek et al., 2015*; *Linsley et al., 2014*; *Pabst et al., 2016*) (CD4 T cells, CD8 T cells, B cells, NK cells, Monocytes and Neutrophils) from a diverse set of patients analyzed in different centers (see Materials and methods). Principal component analysis (PCA) of these data (*Figure 1C*) showed that samples clustered first according to cell type and not according to experiment of origin, patient age, disease status or other factors, suggesting that they could be used as *bona fide* reference expression profiles across different patients. Reference gene expression profiles for each major immune cell type were built from these RNA-Seq samples based on the median normalized counts per gene and cell type. The variability in expression for each gene was also considered when predicting the various cell proportions based on these reference profiles (see Materials and methods and *Supplementary file 1*).

Immune cells differ in their gene expression profiles depending on their state and site of origin (e.g., blood or tumors) (*Ganesan et al., 2017*; *Speiser et al., 2016*; *Zheng et al., 2017*). To study the potential effect of these differences on our predictions, we established reference gene expression profiles of each major tumor-infiltrating immune cell type (i.e., CD4 T, CD8 T, B, NK, macrophages). We further derived reference profiles for stromal cells (i.e. cancer-associated fibroblasts (CAFs)) and endothelial cells. These reference gene expression profiles were obtained as cell type averages from the single-cell RNA-Seq data of melanoma patients from Tirosh and colleagues (*Tirosh et al., 2016*), considering only samples from primary tumor and non-lymphoid tissue metastasis (see Materials and methods and *Supplementary file 2*). As for circulating immune cell data, principal component analysis of the tumor-infiltrating cells' gene expression profiles showed that samples clustered first according to cell type (*Figure 1D* and *Figure 1—figure supplement 1*, see also results in [*Tirosh et al., 2016*]).

## Cancer and non-malignant cell fraction predictions

Reference gene expression profiles from each of the immune and other non-malignant (i.e., stromal and endothelial) cell types were then used to model bulk gene expression data as a linear combination of $m$ different cell types (*Figure 1B*). To include cell types like cancer cells that show high variability across patients and tissues of origin, we further implemented in our algorithm the ability to consider an uncharacterized cell population. Mathematically this was done by taking advantage of the presence of gene markers of non-malignant cells that are not expressed in cancer cells. Importantly, we do not require our signature genes to be expressed in exactly one cell type, but only to show very low expression in cancer cells. The mRNA proportion of each immune and other non-malignant cell type was inferred using least-square regression, solving first our system of equations for the marker genes (green box in *Figure 1B*, see Materials and methods). The fraction of cancer cells was then determined as one minus the fraction of all non-malignant cell types. Cell markers used in this work were determined by differential expression analysis based on our reference cell gene expression profiles as well as gene expression data from non-hematopoietic tissues (see Materials and methods and *Appendix 1—table 1*). Finally, to account for different amounts of mRNA in different cell types and enable meaningful comparison with flow cytometry and IHC data, we measured the mRNA content of all major immune cell types as well as of cancer cells (*Figure 1—figure supplement 2*) and used these values to renormalize our predicted mRNA proportions (see Materials and methods).

## Validation in blood

We first tested our algorithm using three datasets comprising bulk RNA-Seq data from PBMC (*Hoek et al., 2015*; *Zimmermann et al., 2016*) or whole blood (*Linsley et al., 2014*), as well as the corresponding proportions of immune cell types determined by flow cytometry (*Figure 2A*). These data were collected from various cancer-free human donors (see Materials and methods). Overall, very accurate predictions were obtained by fitting reference profiles from circulating immune cells, considering either all cell types together (*Figure 2A*) or each cell type separately (*Figure 2—figure supplement 1*). When comparing with other widely used cell fraction prediction methods (*Becht et al., 2016*; *Gong and Szustakowski, 2013*; *Li et al., 2016*; *Newman et al., 2015*; *Quon et al., 2013*; *Zhong et al., 2013*), we observed a clear improvement (*Figure 2B* and *Figure 2—figure supplement 1*). Of note, the very high correlation values can partly result from the broad range of different cell fractions in our data (*Figure 2A*) and we emphasize that these

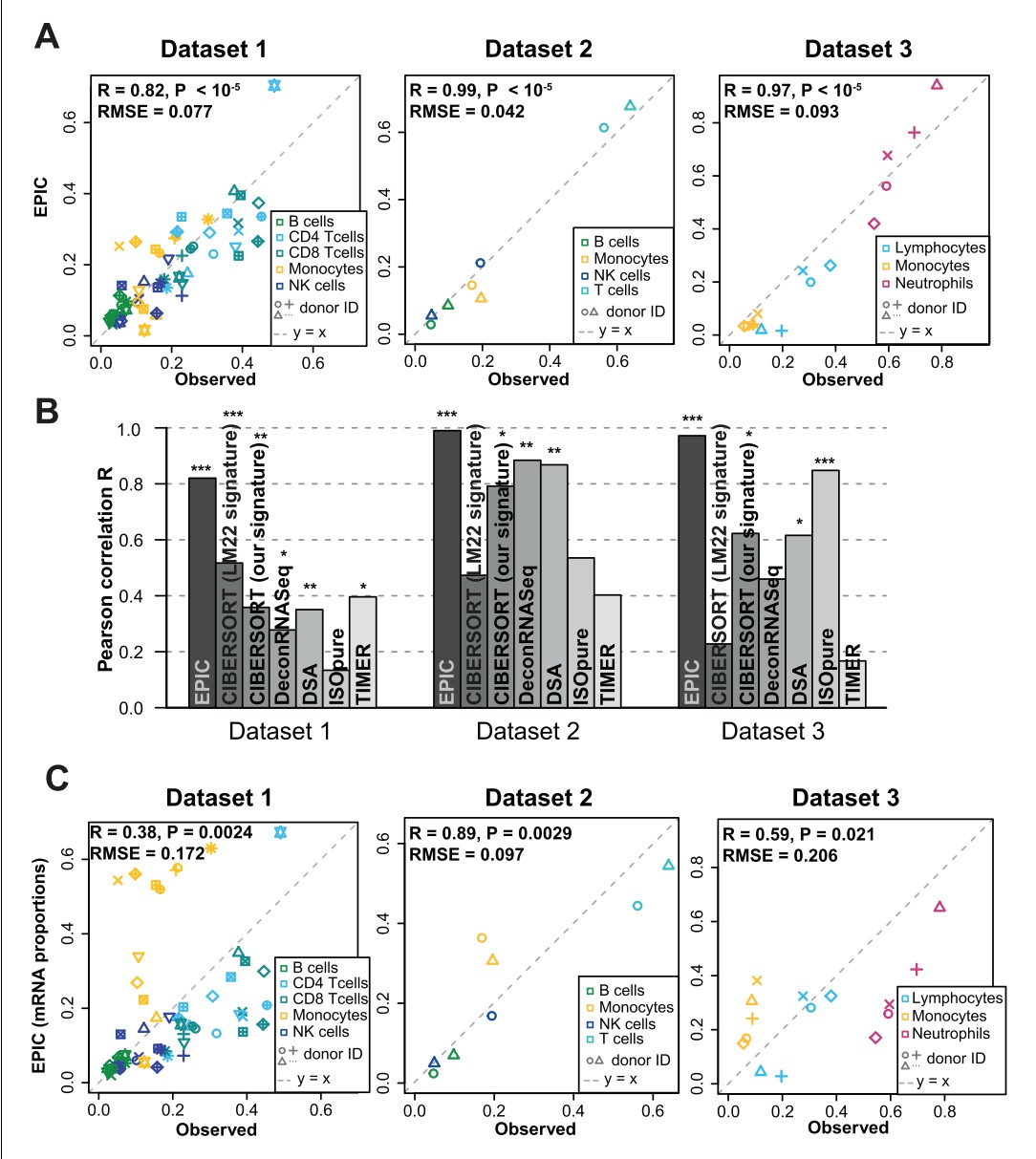

**Figure 2.** Predicting cell fractions in blood samples. (**A**) Predicted vs. measured immune cell proportions in PBMC (dataset 1 (*Zimmermann et al., 2016*), dataset 2 (*Hoek et al., 2015*)) and whole blood (dataset 3 (*Linsley et al., 2014*)); predictions are based on the reference profiles from circulating immune cells. (**B**) Performance comparison with other methods. Significant correlations are indicated above each bar (*p<0.05; **p<0.01; ***p<0.001). (**C**) Predicted immune cells' mRNA proportions (i.e., without mRNA renormalization step) vs. measured values in the same datasets. Correlations are based on Pearson correlation; RMSE: root mean squared error. Proportions of cells observed experimentally are given in *Supplementary file 3B-D*.
DOI: https://doi.org/10.7554/eLife.26476.006

The following figure supplements are available for figure 2:

**Figure supplement 1.** Comparison of multiple cell fraction prediction methods in blood datasets.
DOI: https://doi.org/10.7554/eLife.26476.007

**Figure supplement 2.** Effect of including an mRNA renormalization step for multiple cell fraction prediction methods.
DOI: https://doi.org/10.7554/eLife.26476.008

**Figure supplement 3.** Effect of the various steps in EPIC on the prediction accuracy.
DOI: https://doi.org/10.7554/eLife.26476.009

**Figure supplement 4.** Results with or without known reference profiles for T cells for the cell fraction predictions from various methods.
DOI: https://doi.org/10.7554/eLife.26476.010

correlation values should only be used to compare methods tested on the same datasets (*Figure 2B*). The root mean squared error (RMSE), which is less dependent on such 'outlier data points', shows also improved accuracy for EPIC compared to other methods (*Figure 2—figure supplement 1B*).

The renormalization by mRNA content, which had not been considered in previous approaches, appeared to be important for predicting actual cell fractions (*Figure 2C*). Moreover, we observed that most other methods could also benefit from such a renormalization step, be it for the global prediction of all cell types together or for the predictions of each cell type (*Figure 2—figure supplement 2*). A conceptually related approach was developed by *Baron et al. (2016)*, but the rescaling was done on the gene expression reference profiles (in their case of pancreatic cells) based on the total number of transcripts per cell type and the prediction of the proportion of pancreatic cells from a bulk sample was carried out with CIBERSORT (*Newman et al., 2015*). For comparison purpose, we implemented such an *a priori* renormalization step in EPIC as well, and observed similar results than with the *a posteriori* renormalization (*Figure 2—figure supplement 3*).

With respect to the other methods, EPIC has two other main distinctive features: (i) it allows for a cell type without a known reference gene expression profile (such a feature is also part of ISOpure [*Quon et al., 2013*]) and (ii) it integrates information about the variability in each signature gene from the reference gene expression profiles. The latter point slightly improves the prediction accuracy but to a lesser extent than the renormalization by mRNA (*Figure 2—figure supplement 3*). The former point cannot be tested directly in the blood samples as only cell types with known reference gene expression profiles are composing these samples. Therefore, to test the effect of including a cell type without a known reference profile, we removed all the T cell subsets from the reference gene expression profiles and predicted the proportion of the other cell types in the bulk samples, allowing for one uncharacterized cell type in EPIC. As expected, the results from EPIC including or not the T cell reference profiles remained nearly unchanged, while the other methods suffered from this, except for DSA (*Zhong et al., 2013*) which is based only on signature genes and not reference profiles (*Figure 2—figure supplement 4*). Such an advantage of EPIC is especially useful in the context of tumor samples where in general no reference gene expression profile is available for the cancer cells.

## Validation in solid tumors

To validate our predictions in tumors, we collected single cell suspensions from lymph nodes of four metastatic melanoma patients (see Materials and methods). A fraction of the cell suspension was used to measure the different cell type proportions with flow cytometry (CD4 T, CD8 T, B, NK, melanoma and other cells comprising mostly stromal and endothelial cells; *Supplementary file 3A*), and the other fraction was used for bulk RNA sequencing (*Figure 3—figure supplement 1*). EPIC was first run with reference profiles from circulating immune cells. We observed a remarkable agreement between our predictions and experimentally determined cell fractions (*Figure 3A*). The high correlation value is possibly driven by the two samples containing about 80% of melanoma cells, but we stress that all predicted cell proportions fall nearly on the 'y = x' line. This clearly indicates that the absolute cell fractions were correctly predicted for all cell types, as confirmed by a low RMSE. Of note, the proportion of melanoma cells could be very accurately predicted even in the absence of *a priori* information about their gene expression.

As a second validation, we compared EPIC predictions with IHC data from colon cancer (*Becht et al., 2016*) (see Materials and methods). Although a limited number of immune cell types had been assayed in this study, we observed a significant correlation between cell proportions measured by IHC and our predictions, except for the CD8 T cells (*Figure 3B*).

As a third validation, we used recent single-cell RNA-Seq data from 19 melanoma samples (*Tirosh et al., 2016*). We applied EPIC on the average expression profile over all single cells for each patient and compared the results with the actual cell fractions (see Materials and methods). Here again, our predictions were consistent with the observed cell fractions, even for melanoma cells for which we did not assume any reference gene expression profile (*Figure 3C*). Notably, the predicted cell fractions from melanoma cells as well as from all other immune cell types fall nearly on the y = x line, showing that not only the relative cell type proportions could be predicted but also the absolute proportions for all cell types.

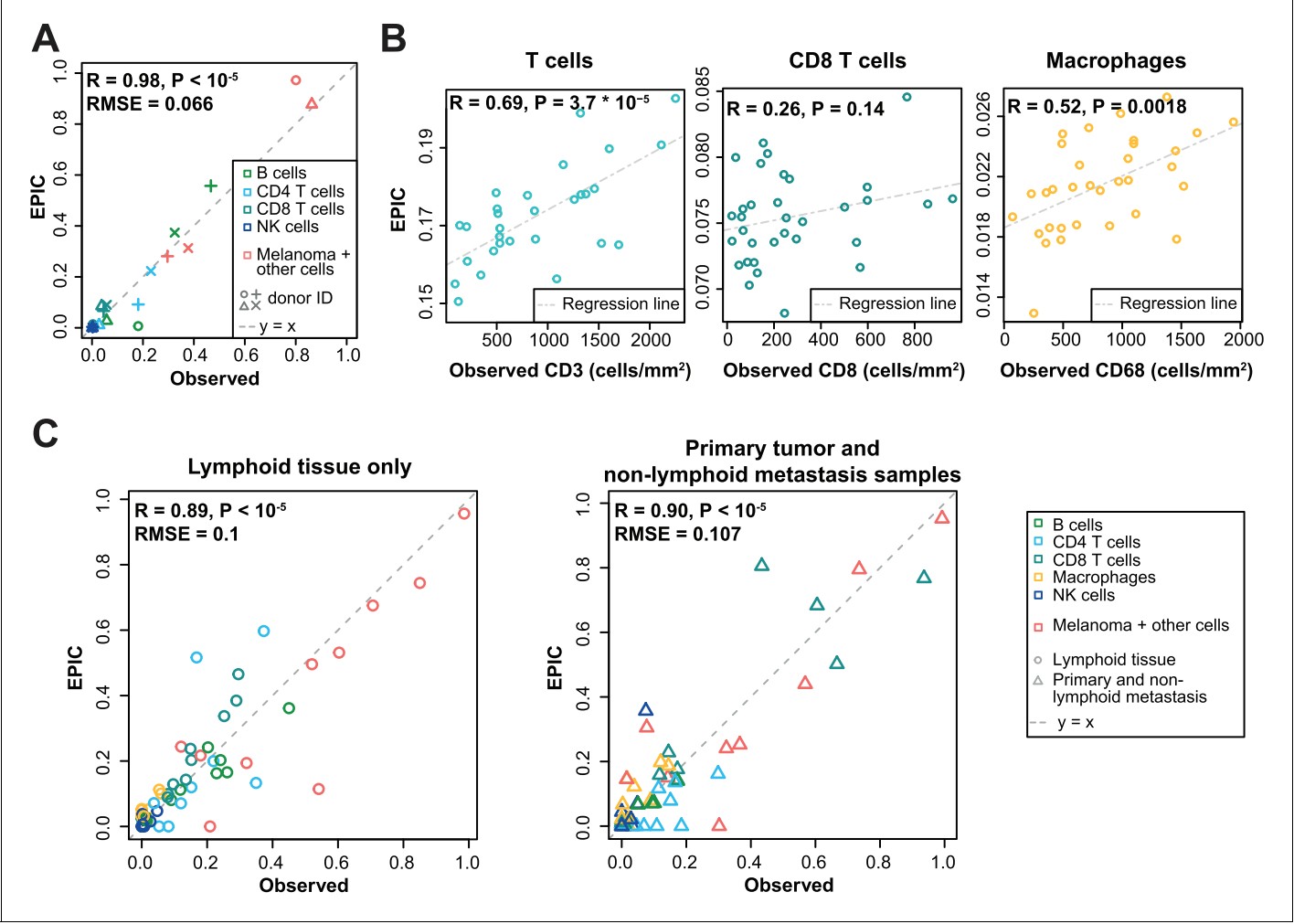

**Figure 3.** Predicting cell fractions in solid tumors with reference profiles from circulating cells. (**A**) Comparison of EPIC predictions with our flow cytometry data of lymph nodes from metastatic melanoma patients. (**B**) Comparison with immunohistochemistry data from colon cancer primary tumors (*Becht et al., 2016*). (**C**) Comparison with single-cell RNA-Seq data (*Tirosh et al., 2016*) from melanoma samples either from lymphoid tissues or primary and non-lymphoid metastatic tumors. Correlations are based on Pearson correlation. Proportions of cells observed experimentally are given in *Supplementary file 3A,E*.

DOI: https://doi.org/10.7554/eLife.26476.011

The following figure supplement is available for figure 3:

**Figure supplement 1.** Sketch of the experiment designed to validate EPIC predictions starting from in vivo tumor samples.

DOI: https://doi.org/10.7554/eLife.26476.012

We next compared these predictions to those obtained with reference profiles from tumor- infil-trating cells, including also CAFs and endothelial cells (*Figure 4*). For the single-cell RNA-Seq data (*Figure 4C*), we applied a leave-one-out procedure, to avoid using the same samples both to build the reference profiles and the bulk RNA-Seq data used as input for the predictions (see Materials and methods). Overall, predictions did not change much compared to those based on circulating immune cell reference gene expression profiles (*Figure 4*), except for the IHC data where the predictions for CD8 T cells and macrophages clearly improved (*Figure 4B*). Moreover, we could observe some differences between the results obtained from circulating immune cell reference gene expression profiles and those from tumor-infiltrating cell reference gene expression profiles, when considering the proportions from each cell type independently (*Figures 3–4* and *Figure 4—figure supplement 1*): (i) predictions for CD8 T cells and macrophages improved in the datasets of primary tumors and non-lymph node metastases but not in the datasets from lymph node metastases; (ii)

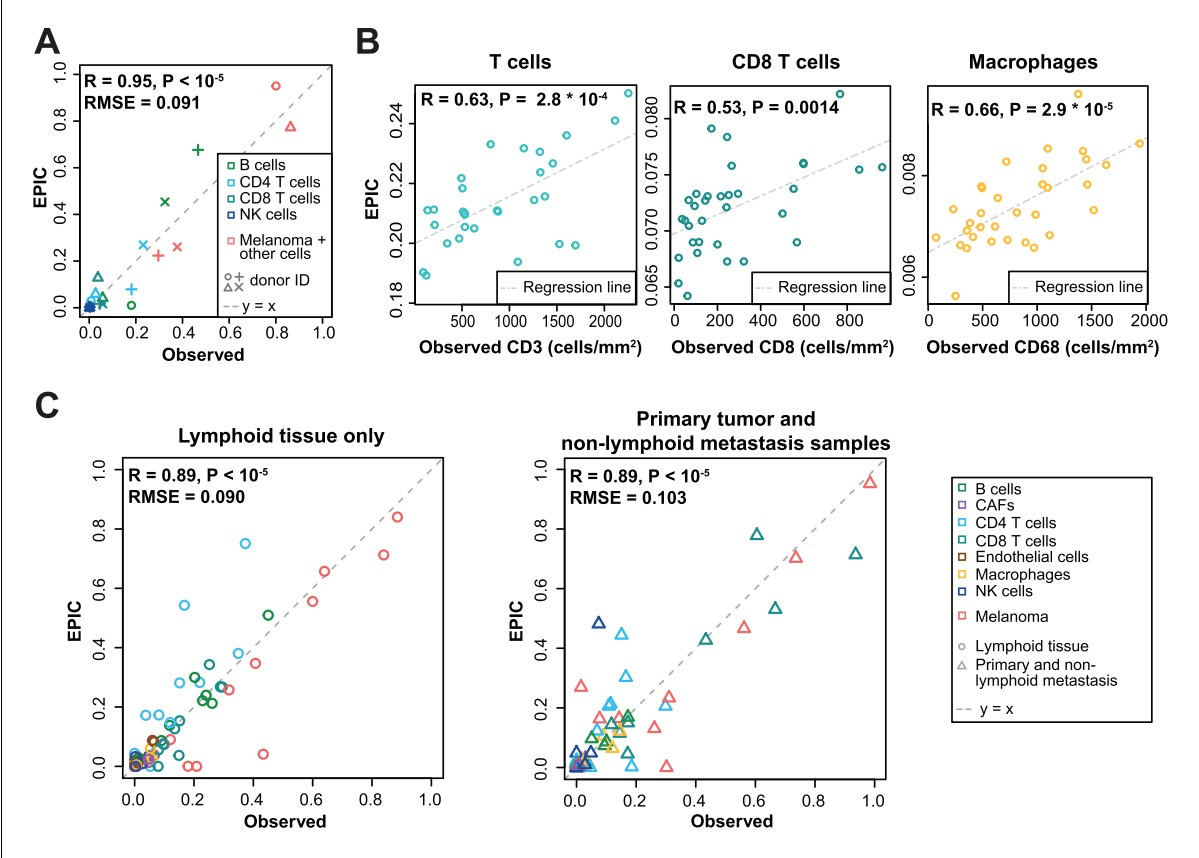

**Figure 4.** Predictions with reference profiles from tumor-infiltrating cells. Same as *Figure 3* but based on reference profiles built from the single-cell RNA-Seq data of primary tumor and non-lymphoid metastatic melanoma samples from *Tirosh et al. (2016)*. (**A**) Comparison with flow cytometry data of lymph nodes from metastatic melanoma patients. (**B**) Comparison with IHC from colon cancer primary tumors (*Becht et al., 2016*). (**C**) Comparison with single-cell RNA-Seq data (*Tirosh et al., 2016*). For primary tumor and non-lymphoid metastasis samples, a leave-one-out procedure was used (see Materials and methods). Proportions of cells observed experimentally are given in *Supplementary file 3A,E*.

DOI: https://doi.org/10.7554/eLife.26476.013

The following figure supplement is available for figure 4:

**Figure supplement 1.** Comparison of EPIC results per cell type for gene expression reference profiles from circulating or tumor-infiltrating immune cells.

DOI: https://doi.org/10.7554/eLife.26476.014

predictions for B and NK cells displayed similar accuracy based on the circulating cells or tumor-infiltrating cells profiles for all datasets; and (iii) CAFs and endothelial cells were not available for the blood-based reference profiles but the proportion of these cells could be well predicted based on the reference profiles constructed from tumor-infiltrating cells (*Figure 4* and *Figure 4—figure supplement 1*).

## Benchmarking of other methods

We took advantage of our unique collection of independent validation datasets to benchmark other methods for predictions of immune cell type fractions in human tumors. We first compared the results of EPIC and ISOpure (*Quon et al., 2013*), which is the only other method that can consider uncharacterized cell types and therefore predict the fraction of cancer and immune cell types based only on RNA-seq data. EPIC displayed improved accuracy in all three datasets (*Figure 5A*, and *Figure 5—figure supplements 1–4*). To benchmark other methods, we then restricted our analysis to the predictions of the different immune cell types (*Figure 5B* and *Figure 5—figure supplements 1–4*). Predictions from EPIC were in general more accurate, especially when considering all cell types

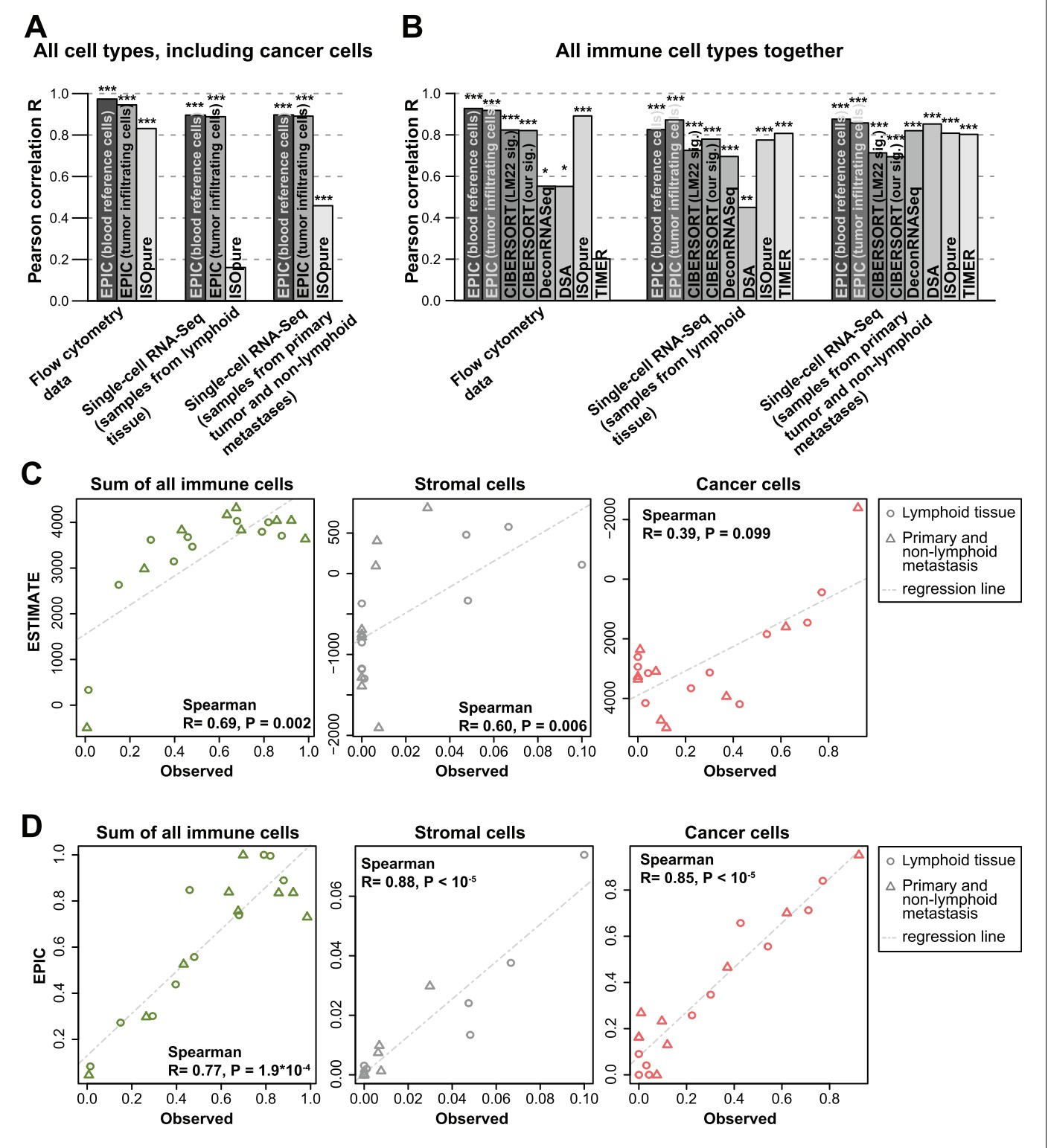

**Figure 5.** Performance comparison with other methods in tumor samples. (**A**) Pearson correlation R-values between the cell proportions predicted by EPIC and ISOpure and the observed proportions measured by flow cytometry or single-cell RNA-Seq (*Tirosh et al., 2016*), considering all cell types together (i.e., B, CAFs, CD4 T, CD8 T, endothelial, NK, macrophages and cancer cells). (**B**) Same analysis as in *Figure 5A* but considering only immune cell types (i.e., B, CD4 T, CD8 T, NK and macrophages) in order to include more methods in the comparison. (**C**) Analysis of ESTIMATE predictions in the single-cell RNA-Seq dataset for the sum of all immune cells, the proportion of stromal cells (cancer-associated fibroblasts) and the proportion of

*Figure 5 continued on next page*

*Figure 5 continued*

cancer cells (cells identified as melanoma cells in Tirosh et al.). (D) Same as *Figure 5C* but for EPIC predictions of immune, stromal and cancer cells. Significant correlations in (A–B) are indicated above each bar (*p<0.05; **p<0.01; ***p<0.001).

DOI: https://doi.org/10.7554/eLife.26476.015

The following figure supplements are available for figure 5:

**Figure supplement 1.** Comparison of multiple cell fraction prediction methods in tumor datasets.

DOI: https://doi.org/10.7554/eLife.26476.016

**Figure supplement 2.** Comparison of cell fraction prediction methods with flow cytometry data of melanoma tumors.

DOI: https://doi.org/10.7554/eLife.26476.017

**Figure supplement 3.** Comparison of cell fraction prediction methods with immunohistochemistry data in colon cancer data (*Becht et al., 2016*) for T cell, CD8 T cell and macrophage infiltration values.

DOI: https://doi.org/10.7554/eLife.26476.018

**Figure supplement 4.** Comparison of cell fraction prediction methods with single-cell RNA-Seq data from melanoma tumors (*Tirosh et al., 2016*).

DOI: https://doi.org/10.7554/eLife.26476.019

**Figure supplement 5.** Comparison between ESTIMATE scores (A) and EPIC predictions (B) in our new flow cytometry dataset.

DOI: https://doi.org/10.7554/eLife.26476.020

**Figure supplement 6.** Predicting Thelper and Treg cell fractions in tumors.

DOI: https://doi.org/10.7554/eLife.26476.021

together. Nevertheless, when restricting the comparisons to relative cell type proportions, some methods like MCPcounter (*Becht et al., 2016*) and TIMER (*Li et al., 2016*) were quite consistent in their predictions across the various datasets and showed similar accuracy as EPIC for the cell types considered in these methods (*Figure 5—figure supplements 1–4*). Of note, MCPcounter could not be included in the global prediction comparison as this method returns scores that are not comparable between different cell types. Predictions from DSA (*Zhong et al., 2013*) were also quite accurate when available, but in multiple cases some cell type proportions returned by the method were simply equal to 0 in all samples (*Figure 5—figure supplements 1–4* – correlation values were replaced by *NA* in these cases).

Immune, stromal and tumor purity scores (*Yoshihara et al., 2013*) based on gene set enrichment analysis were also correlated with the total fraction of immune cells, stromal cells and the fraction of cancer cells (correlation was not significant for the tumor purity score – *Figure 5C* and *Figure 5—figure supplement 5*). However, these correlations were considerably lower than those obtained with our approach (*Figure 5D* and *Figure 5—figure supplement 5*). Moreover, such scores are less quantitative and are thus more difficult to interpret with respect to actual cell type proportions.

Next, we performed some explorative analysis to test if such cell fraction prediction methods could go further into the details of the T cell subsets. Based on the single-cell RNA-seq data from Tirosh and colleagues (*Tirosh et al., 2016*), we identified CD4+ Treg and Thelper cells (see Materials and methods), and built reference gene expression profiles for these cell types as we had done for the other cell types. As before, a leave-1-out procedure at the patient level was used to predict the proportions for these cell types based on the bulk samples constructed from this single-cell RNA-seq data. We observed significant correlations, but the R-values are lower than before (*Figure 5—figure supplement 6*), suggesting that we may be reaching the limits of cell fraction prediction methods for cell types that show lower abundance and substantial transcriptional similarity. Of note, CIBERSORT using either the LM22 signature or our newly derived signature did not perform better than EPIC.

## Discussion

By combining RNA-Seq profiles of all major immune and other non-malignant cell types established from both circulating and tumor-infiltrating cells together with information about cell morphology (i.e., mRNA content) and algorithmic developments to consider uncharacterized and possibly highly variable cell types, EPIC overcomes several limitations of previous approaches to predict the fraction of both cancer and immune or other non-malignant cell types from bulk tumor gene expression data. From an algorithmic point of view, EPIC takes advantage of the fact that cancer cells, in general, express no or only low levels of immune and stromal markers. Therefore the method can be

broadly applied to most solid tumors, as confirmed by our validation in melanoma and colorectal samples, but it will not be suitable for hematological malignancies like leukemia or lymphoma.

The accuracy of the predictions for some cell types might be sensitive to the origin or condition of the immune cells used for establishing reference profiles. For instance, we observed that CD8 T cells and macrophages from primary tumors and non-lymph node metastases samples were best predicted using the reference profiles from tumor-infiltrating cells. This may be explained by the fact that the reference profiles from circulating cells corresponded to resting CD8 T cells and monocytes as only few activated C8 T cells and no macrophages are circulating in blood.

Overall, our results suggest that for primary tumors or non-lymphoid tissue metastases reference gene expression profiles from tumor-infiltrating immune cells are more appropriate, while for lymph node metastases, profiles from either circulating or tumor-infiltrating immune cells could be used.

One known limitation of cell fraction predictions arises when some cell types are present at very low frequency (*Shen-Orr and Gaujoux, 2013*). Our results suggest that predictions of cell proportions are reliable within an absolute error of about 8%, as estimated by the root mean squared error (*Figure 3* and *Figure 4—figure supplement 1B*). These estimates are consistent with the lower detection limit proposed by other groups (*Becht et al., 2016*; *Zhong et al., 2013*) and may explain why the relative proportions of NK cells, which are present at lower frequency in melanoma tumors (*Balch et al., 1990*; *Sconocchia et al., 2012*), could not be predicted with accuracy comparable to other cell types (*Figure 4—figure supplement 1*). While this may prevent applications of cell fraction predictions in some tumor types that are poorly infiltrated, many other tumors, like melanoma or colorectal cancer, display high level of infiltrating immune cells and the role of immune infiltrations on tumor progression and survival appears to be especially important in these tumors (*Clemente et al., 1996*; *Fridman et al., 2012*; *Galon et al., 2006*).

Another limitation of cell fraction predictions arises when considering cell types that show high transcriptional similarity. For instance, Treg in *Figure 5—figure supplement 6* could not be well predicted neither by EPIC nor CIBERSORT. This can be understood because the gene expression profiles of Thelper and Treg are highly similar with only a few genes expressed differently between the two, making it harder for cell fraction prediction methods to work accurately. In addition, Treg were present at a proportion lower than 10% in all samples (*Supplementary file 3E*). For such cases, gene set enrichment approaches, although less quantitative, may be more convenient, possibly combining them with the predictions obtained by EPIC for the main immune cell types.

Our predictions for the fraction of uncharacterized cells may include non-malignant cells, such as epithelial cells from neighboring tissues in addition to cancer cells. Compared to recent algorithms that first predict tumor purity based on exome sequencing data, and later infer the relative fraction of immune cell types (*Li et al., 2016*), the predictions of EPIC for immune and stromal cells are likely more quantitative because they can implicitly consider the presence of not only cancer cells but also other non-malignant cells for which reference profiles are not available. Moreover, EPIC does not require both exome and RNA-Seq data from the same tumor samples, thereby reducing the cost and amount of experimental work for prospective studies, and broadening the scope of retrospective analyses of cancer genomics data to studies that only include gene expression data.

Recent technical developments in single-cell RNA-Seq technology enable researchers to directly access information about both the proportion of all cell types together with their gene expression characteristics (*Carmona et al., 2017*; *Efroni et al., 2015*; *Jaitin et al., 2014*; *Singer et al., 2016*; *Stegle et al., 2015*; *Tirosh et al., 2016*). Such data are much richer than anything that can be obtained with computational deconvolution of bulk gene expression profiles and this technology may eventually replace standard gene expression analysis of bulk tumors. Nevertheless, it is important to realize that, even when disregarding the financial aspects, single-cell RNA-Seq of human tumors is still logistically and technically very challenging due to high level of cell death upon sample manipulation (especially freezing and thawing) and high transcript dropout rates (*Finak et al., 2015*; *Saliba et al., 2014*; *Stegle et al., 2015*). Moreover, one cannot exclude that some cells may better survive the processing with microfluidics devices used in some single-cell RNA-Seq platforms, thereby biasing the estimates of cell type proportions. It is therefore likely that bulk tumor gene expression analysis will remain widely used for several years. Our work shows how we can exploit recent single-cell RNA-Seq data of tumors obtained from a few patients to refine cell fraction predictions in other patients that could not be analyzed with this technology, thereby overcoming some

limitations of previous computational approaches that relied only on reference gene expression profiles from circulating immune cells.

In this work, we provide a detailed and biologically relevant validation of our predictions using actual tumor samples from human patients analyzed with flow cytometry, IHC and single-cell RNA-Seq. We note that the slightly lower agreement between our predictions and IHC data may be partly explained by the fact that the exact same samples could not be used for both gene expression and IHC analyses because of the incompatibility between the two techniques. Nevertheless, the overall high accuracy of our predictions indicates that infiltrations of major immune cell types can be quantitatively studied directly from bulk tumor gene expression data using computational approaches such as EPIC.

EPIC can be downloaded as a standalone R package (https://github.com/GfellerLab/EPIC/releases/tag/v1.1) (*Racle, 2017*) and can be used with reference gene expression profiles pre-compiled from circulating or tumor-infiltrating cells, or provided by the user. EPIC is also available as a web application (http://epic.gfellerlab.org) where users can upload bulk samples gene expression data and perform the full analysis.

## Materials and methods

### Code availability

EPIC has been written as an R package. It is freely available on GitHub (https://github.com/GfellerLab/EPIC; copy archived on https://github.com/elifesciences-publications/EPIC) for academic non-commercial research purposes. Version v1.1 of the package was used for our analyses.

In addition to the R package, EPIC is available as a web application at the address: http://epic.gfellerlab.org.

### Prediction of cancer and immune cell type proportions

In EPIC, the gene expression of a bulk sample is modeled as the sum of the gene expression profiles from the pure cell types composing this sample (*Figure 1A,B*). This can be written as (*Venet et al., 2001*):

$$b = C \times p \tag{1}$$

Where $b$ is the vector of all $n$ genes expressed from the bulk sample to deconvolve; $C$ is a matrix ($n \times m$) of the $m$ gene expression profiles from the different cell types; and $p$ is a vector of the proportions from the $m$ cell types in the given sample (*Figure 1B*).

Matrix $C$ consists of $m-1$ columns corresponding to various reference non-malignant cell types whose gene expression profiles are known, and a last column corresponding to uncharacterized cells (i.e. mostly cancer cells, but possibly also other non-malignant cell types not included in the reference profiles). EPIC assumes the reference gene expression profiles from the non-malignant cell types are well conserved between patients. Such a hypothesis is supported by the analysis in *Figure 1C,D*. The uncharacterized cells can be more heterogeneous between patients and EPIC makes no assumption on them.

EPIC works with RNA-seq data, which is implicitly normalized. We use data normalized into transcripts per million (TPM) because it has some properties needed for the full cell proportion prediction to work, as will be shown in the next paragraphs. Therefore, instead of the raw data from *Equation (1)*, we are working with TPM-normalized data, which correspond to the following:

$$\bar{b}_i = \frac{10^6}{\sum_{k=1}^{n} \frac{b_k}{l_k}} \cdot \frac{b_i}{l_i}$$

$$\bar{C}_{ij} = \frac{10^6}{\sum_{k=1}^{n} \frac{C_{kj}}{l_k}} \cdot \frac{C_{ij}}{l_i} \tag{2}$$

Where $\bar{b}$ and $\bar{C}$ are the TPM-normalized bulk sample and reference cell gene expression profiles respectively and $l_i$ is the length of gene $i$.

Using these, *Equation (1)* is rewritten to:

$$\bar{b} = \bar{C} \times \bar{p} \qquad (3)$$

where

$$\bar{p}_j = \frac{\sum_{k=1}^{n} \frac{C_{kj}}{l_k}}{\sum_{i=1}^{n} \frac{b_i}{l_i}} \cdot p_j \qquad (4)$$

This normalization guarantees the sum of the new proportions, $\bar{p}$, is equal to 1:

$$\sum_{i=1}^{n} \bar{b}_i \overset{\text{from eq. 2}}{=} \sum_i \left( \frac{10^6}{\sum_k \frac{b_k}{l_k}} \cdot \frac{b_i}{l_i} \right) = 10^6$$

and

$$\sum_{i=1}^{n} \bar{b}_i \overset{\text{from eq. 3}}{=} \sum_i (\bar{C} \times \bar{p})_i = \sum_{i=1}^{n} \sum_{j=1}^{m} \overline{C_{ij}} \cdot \bar{p}_j \overset{\text{from eq. 2}}{=} \sum_j \left[ \left( \frac{10^6}{\sum_k \frac{C_{kj}}{l_k}} \cdot \sum_i \frac{C_{ij}}{l_i} \right) \cdot \bar{p}_j \right] = 10^6 \cdot \sum_j \bar{p}_j$$

$$\Rightarrow \sum_{j=1}^{m} \bar{p}_j = 1 \qquad (5)$$

In addition to $\bar{p}$ and $\bar{C}$ we also define $\bar{p}^*$ and $\bar{C}^*$, which are the same except that they include the normalized proportions and profiles from the reference cell types only (i.e. they have one less element and one less column than $\bar{p}$ and $\bar{C}$ respectively).

Using these normalized quantities, EPIC then solves *Equation (3)* for a subset of $n_s$ equations corresponding to the $n_s$ signature genes (S) that are expressed by one or more of the reference cell types but only expressed at a negligible level in the uncharacterized cells (*Figure 1B*). Previous computational work (*Clarke et al., 2010*; *Gosink et al., 2007*) showed that the proportion from uncharacterized cells in bulk samples could indeed be inferred with help of genes not expressed by the uncharacterized cells. Importantly, cell-specific signature genes are well established and widely used in flow cytometry to sort immune cells. Thus, EPIC solves the following system of equations:

$$\bar{b}_i|_{i \in S} = ((\bar{C}^* \times \bar{p}^*)_i + \bar{C}_{im} \cdot \bar{p}_m)|_{i \in S} = (\bar{C}^* \times \bar{p}^*)_i|_{i \in S} \qquad (6)$$

where the term corresponding to the uncharacterized cells (m) vanished thanks to the definition of the signature genes ($\bar{C}_{im}|_{i \in S} \cong 0$).

The solution to *Equation (6)* can be estimated by a constrained least square optimization. EPIC takes advantage of the known variability for each gene in the reference profile, to give weights in the function to minimize:

$$f(\bar{p}^*) = \sum_{i \in S} w_i \left[ \bar{b}_i - (\bar{C}^* \times \bar{p}^*)_i \right]^2 \qquad (7)$$

with constraints

$$\begin{cases} \bar{p}_j^* \geq 0 \\ \sum_{j=1}^{m-1} \bar{p}_j^* \leq 1 \end{cases}$$

Here, the weights, $w_i$, give more importance to the signature genes with low variability in the reference gene expression profiles. In EPIC, these weights are given by:

$$w_i = \min(u_i, \ 100 \cdot \text{median}(u_i))$$

where

$$u_i = \sum_{j=1}^{m-1} \frac{\bar{C}_{ij}^*}{\bar{V}_{ij} + \varepsilon}$$

$\bar{V}$ is the matrix of the TPM-based variability of each gene for each of the reference cells (*Supplementary files 1–2*), $\varepsilon$ is a small number to avoid division by 0, and the term '$100 \cdot \mathrm{median}(u_i)$' is used to avoid giving too much weight to few of the genes.

Finally, thanks to *Equation (5)*, the proportion for the uncharacterized cells can be obtained by:

$$\bar{p}_m = 1 - \sum_{j=1}^{m-1} \bar{p}_j \qquad (8)$$

Since we used normalized gene expression data, values of $\bar{p}$ correspond actually to the fraction of mRNA coming from each cell type, rather than the cell proportions. As the mRNA content per cell type can vary significantly (*Figure 1—figure supplement 2*), the actual proportions of each cell type can be estimated as:

$$p_j = \alpha \cdot \frac{\bar{p}_j}{r_j} \qquad (9)$$

where $r_j$ is the amount of RNA nucleotides in cell type $j$ (or equivalently the total weight of RNA in each cell type) and $\alpha$ is a normalization constant to have $\sum p_j = 1$.

Another method, DeMix (*Ahn et al., 2013*), starts from *Equation (1)* to estimate the proportion and gene expression profile from cancer cells in mixed samples. In this method the mixed sample is assumed to be composed only by two cell types: cancer cells (without any *a priori* known gene expression profile) and normal cells (with known gene expression data, which can either come from tumor-matched or unmatched samples). This method was developed for microarray data and shows that it was important to use the raw data as input assuming it follows a log-normal distribution as is the case for microarray, instead of working with log-transformed data like most other methods did. DeMix estimates the variance of the gene expression in the normal samples and uses this in the maximum likelihood estimation to predict the cancer cell gene expression and proportions, using thus implicitly a gene-specific weight for each gene. EPIC was derived for RNA-seq data and is directly using linear (non-log) data. Notably, when solving the linear regression from *Equation (7)* in EPIC, we are not assuming any specific distribution for the gene expression and the weights we give to each gene are simply based on their interquartile range in the reference samples. Other measures of gene variability could however be given as input into EPIC's *R-package*. Contrary to DeMix, another important assumption performed in EPIC is that our signature genes are not expressed by the cancer cells, so that we can easily estimate the proportion of multiple non-malignant cell types composing the bulk samples.

## Flow cytometry and gene expression analysis of melanoma samples

Patients agreed to donate metastatic tissues upon informed consent, based on dedicated clinical investigation protocols established according to the relevant regulatory standards. The protocols were approved by the local IRB, that is, the 'Commission cantonale d'éthique de la recherche sur l'être humain du Canton de Vaud'. Lymph nodes (LN) were obtained from stage III melanoma patients, by lymph node dissection that took place before systemic treatment. The LN from one patient was from the right axilla and the LNs from the other three patients were from the iliac and ilio-obturator regions (*Appendix 2—table 1*). Single cell suspensions were obtained by mechanical disruption and immediately cryopreserved in RPMI 1640 supplemented with 40% FCS and 10% DMSO. Single cell suspensions from four lymph nodes were thawed and used in parallel experiments of flow cytometry and RNA extraction. In order to limit the number of dead cells after thawing, we removed those cells using a dead cell removal kit (Miltenyi Biotech). Proportions of CD4 T (CD45$^+$/CD3$^+$/CD4$^+$/Melan-A$^-$), CD8 T (CD45$^+$/CD3$^+$/CD8$^+$/Melan-A$^-$), NK (CD45$^+$/CD56$^+$/CD3$^-$/CD33$^-$/Melan-A$^-$), B (CD45$^+$/CD19$^+$/CD3$^-$/CD33$^-$/Melan-A$^-$) and Melan-A expressing tumor cells (*Supplementary file 3A*) were acquired via flow cytometry using the following antibodies: anti-CD3 BV711 (RRID:AB_2566035), anti-CD4 BUV737 (RRID:AB_2713927), anti-CD8 PE-Cy5 (RRID:AB_2713928), anti-CD56 BV421 (RRID:AB_11218798), anti-CD19 APCH7 (RRID:AB_1645728), anti-CD33 PE-Cy7 (RRID:AB_2713932), anti-CD45 APC (RRID:AB_314400), anti-Melan-A FITC (RRID:AB_2713930) and Fixable Viability Dye eFluor 455UV (eBioscience). Data was acquired on a BD LSR II SORP flow cytometry machine (BD Bioscience). Analysis was performed using FlowJo (Tree Star).

Cell proportions were based on viable cells only. In parallel total RNA was extracted using the RNAeasy Plus mini kit (Qiagen) following the manufactures' protocol. Starting material always contained minimum $0.2 \times 10^6$ cells. RNA was quantified and integrity was analyzed using a Fragment Analyser (Advanced Analytical). Total RNA from all samples used for sequencing had an RQN $\geq 7$. Libraries were obtained using the Truseq stranded RNA kit (Illumina). Single read (100 bp) was performed using an Illumina HiSeq 2500 sequencer (Illumina).

Post processing of the sequencing was done using Illumina pipeline Casava 1.82. FastQC (version 0.10.1) was used for quality control. RNA-seq reads alignment to the human genome, *hg19*, and TPM quantification were performed with *RSEM* (*Li and Dewey, 2011*) version 1.2.19, using *Bowtie2* (*Langmead and Salzberg, 2012*) version 2.2.4 and *Samtools* (*Li et al., 2009*) version 1.2.

RNA-Seq data from this experiment have been deposited in NCBI's Gene Expression Omnibus (*Edgar et al., 2002*) and are accessible through GEO Series accession number GSE93722 (https://www.ncbi.nlm.nih.gov/geo/query/acc.cgi?acc=GSE93722).

## Amount of mRNA per cell type

Healthy donor peripheral blood was obtained through the blood transfusion center in Lausanne. PBMCs were purified by density gradient using Lymphoprep (Axis-Shieldy). Mononuclear cells were stained in order to sort monocytes, B, T and NK cells using the following antibodies: CD14 FITC (RRID:AB_130992), CD19 PE (RRID:AB_2716572), CD3 APC (RRID:AB_130788), CD56 BV421 (RRID: AB_11218798) and fixable live/dead near IR stain (ThermoFisher Scientific). $1 \times 10^6$ live cells from each cell type were sorted using the BD FACS ARIA III (BD Biosciences). Total RNA was extracted using the RNAeasy Plus mini kit (Qiagen) following the manufactures' protocol and quantified using a Fragment Analyser (Advanced Analytical). Values obtained are given in *Figure 1—figure supplement 2A*.

The mRNA content for the cancer cells was estimated from the flow cytometry data described in the previous section from the four patients with melanoma. For this we used the forward scatter width, which is a good proxy of cell size and mRNA content (*Padovan-Merhar et al., 2015*; *Tzur et al., 2011*), and observed that cancer cells had similar amount of mRNA than B, NK and T cells (*Figure 1—figure supplement 2B*). We thus used a value of 0.4 pg of mRNA per cancer cell.

## Public external datasets used in this study

- Dataset 1 was obtained from Zimmermann and colleagues (*Zimmermann et al., 2016*), through ImmPort (http://www.immport.org), accession SDY67. It includes RNA-Seq samples from PBMC of healthy donors before and after influenza vaccination. In addition, the original flow cytometry results files were available, containing multiple immune cell markers. As an independent validation of EPIC, we used the data from 12 pre-vaccination samples of healthy donors and we computed the corresponding immune cell proportions from the flow cytometry files merging the results from their *innate* and *Treg panels* (obtaining B, CD4 T, CD8 T, NK cells and monocytes, *Supplementary file 3B*; *Figure 2A*)
- Dataset 2 was obtained from Hoek and colleagues (*Hoek et al., 2015*), GEO accession GSE64655. This corresponds to RNA-Seq samples from two different donors. Samples have been taken before an influenza vaccination and also 1, 3 and 7 days after the vaccination (56 samples in total). In their experiment, the authors measured RNA-Seq from bulk PBMC samples and also from sorted immune cells (B, NK, T cells, myeloid dendritic cells, monocytes and neutrophils). In addition, flow cytometry was performed to measure the proportion of these cell types in PBMC before the influenza vaccination (personally communicated by the authors; *Supplementary file 3C*).
- Dataset 3 was obtained from Linsley and colleagues (*Linsley et al., 2014*), GEO accession GSE60424. This dataset includes 20 donors (healthy donors and other donors with amyotrophic lateral sclerosis, multiple sclerosis, type one diabetes or sepsis), for a total of 134 samples. RNA-Seq from these donors has been extracted from whole blood and sorted immune cells (B, NK cells, monocytes, neutrophils, CD4 T and CD8 T cells). In addition to RNA-Seq data, complete blood counts data were available for 5 of these donors (personally communicated by the authors; *Supplementary file 3D*).
- Dataset 4 was obtained from Pabst and colleagues (*Pabst et al., 2016*), GEO accession GSE51984. This includes RNA-Seq from five healthy donors. Samples are from total white blood cells and sorted immune cells (B cells, granulocytes, monocytes and T cells).

- Colon cancer dataset was obtained from Becht and colleagues (*Becht et al., 2016*), GEO accession GSE39582. This corresponds to microarrays of primary colon cancer tumors. In addition to gene expression data, immunohistochemistry data of CD3, CD8 and CD68 was available for 33 patients (personally communicated by the authors).
- Single-cell RNA-Seq data from tumor-infiltrating cells were obtained from Tirosh and colleagues (*Tirosh et al., 2016*), GEO accession GSE72056. This corresponds to 19 donors and comprises primary tumors, lymph node metastasis or other lesions. It includes 4645 cells. Cell type identity was taken from *Tirosh et al. (2016)* (B, NK, T cell, macrophage, CAFs, endothelial cell, cancer cell as well as cells not assigned a specific cell type). Among the T cells, we then defined subsets based on their gene expression: CD4 T cells (expressing CD4 but not CD8A nor CD8B) and CD8 T cells (expressing CD8A or CD8B but not CD4). The other T cells not corresponding to one of these two groups were removed from further analyses. We further defined among the CD4 T cells: Treg (expressing either FOXP3 or CD25 above the median among CD4 T cells) and Thelper (those CD4 T cells not belonging to Treg group). In silico reconstructed bulk samples from each donor were obtained as the average per gene from all samples of the given donor. The corresponding cell fractions from these bulk samples were obtained as the number of cells from each cell type divided by the total number of cells (*Supplementary file 3E*). In the results, we split this dataset in two depending on the origin of the biopsies: lymphoid tissues for samples obtained from lymph node and spleen metastases, vs. the rest of samples, which were obtained from primary tumor and other metastases.

For the above datasets 2 and 3, we obtained raw *fastq* files. RNA-seq reads alignment to the human genome, *hg19*, and TPM quantification were performed with *RSEM* (*Li and Dewey, 2011*) version 1.2.19, using *Bowtie2* (*Langmead and Salzberg, 2012*) version 2.2.4 and *Samtools* (*Li et al., 2009*) version 1.2.

For the other datasets, we directly obtained the summary counts data from the respective studies without mapping the reads by ourselves, and we transformed these counts to TPM wherever necessary.

## Reference gene expression profiles from circulating cells

Reference gene expression profiles of sorted immune cells from peripheral blood were built from the datasets 2, 3 and 4 described in previous section. We verified no experimental biases were present in these data through unsupervised clustering of the samples, with help of a principal component analysis based on the normalized expression from the 1000 most variable genes (*Figure 1C*).

The median value of TPM counts was computed per cell type and per gene. Similarly, the interquartile range of the TPM counts was computed per cell type and gene, as a measure of the variability of each gene expression in each cell type. Values of these reference profiles are given in *Supplementary file 1*). Granulocytes from dataset 4 and neutrophils from datasets 2 and 3 were combined to build the reference profile for neutrophils (neutrophils constitute more than 90% of granulocytes). No reference profile was built for the myeloid dendritic cells as only few samples of these sorted cells existed and they were all from the same experiment. Monocytes are not found in tumors but instead there are macrophages, mostly from monocytic lineage, that are infiltrating tumors and that are not found in blood. For this reason, we also used the monocyte reference gene expression profile as a proxy for macrophages when applying EPIC to tumor samples. Such an assumption gave coherent results as observed in the results.

## Reference profiles from tumor-infiltrating cells

We also built gene expression reference profiles from tumor-infiltrating cells. These are based on the single-cell RNA-Seq data from Tirosh and colleagues (*Tirosh et al., 2016*) described above. We only used the non-lymphoid tissue samples to build these tumor-infiltrating cell's profiles, avoiding in this way potential 'normal immune cells' present in the lymph nodes and spleen. These reference profiles (*Supplementary file 2*) were built in the same way as described above for the reference profiles of circulating immune cells, but based on the mean and standard deviation instead of median and interquartile range respectively, due to the nature of single-cell RNA-Seq data and gene dropout present with such technique.

When testing EPIC with these profiles for the single-cell RNA-Seq datasets, for the samples of primary tumor and other non-lymph node metastases, a leave-one-out procedure was applied: for each donor we built reference cell profiles based only on the data coming from the other donors.

## Cell marker gene identification

EPIC relies on signature genes that are expressed by the reference cells but not by the uncharacterized cells (e.g., cancer cells). For each reference cell type, we thus built a list of signature genes through the following steps:

1. The samples (from the peripheral blood datasets) from each immune cell type were tested for overexpression against:
   a. the samples from each other immune cell of peripheral blood datasets (1 cell type vs. 1 other cell type at a time);
   b. the samples of the Illumina Human Body Map 2.0 Project (ArrayExpress ID: E-MTAB-513) considering all non-immune related tissues;
   c. the samples from GTEx (*GTEx Consortium, 2015*) from each of the following tissues (one tissue at a time): adipose subcutaneous; bladder; colon-transverse; ovary; pancreas; testis (data version V6p).
2. Only genes overexpressed in the given cell type with an adjusted p-value<0.01 for all these tests were kept. Conditions 1b) and 1c) are there to ensure signature genes are expressed in the immune cells and no other tissues.
3. The genes that passed 2) were then ranked by the fold change from the overexpression tests to preselect the genes showing the biggest difference between the various cell types.
4. The list of genes was then manually curated, comparing the expression of the genes per cell type in the peripheral blood datasets and the tumor-infiltrating cells dataset: only genes expressed at similar levels between the blood and tumor-infiltrating cells were kept (to avoid differences due to exhaustion phenotype for example). Genes expressed at much higher levels than the other genes were also removed from the signature as these could have biased the least-square optimization towards good predictions for these genes only.
5. In addition to CD4 and CD8 T cells signatures, we built a signature list of general T cell genes in the same way as described above, and it contains genes expressed at similar levels in the two T cell subtypes. This general T cell signature is also part of EPIC, even when predicting the proportions of the T cell subsets.
6. For the tumor-infiltrating cell reference profiles, signature genes from CAFs, endothelial cells and macrophages were also needed. These were built in a similar way than above, considering the overexpression from each of these cell types against each other cell type from the tumor-infiltrating cells data.

All the differential expression tests were performed with *DESeq2* (*Love et al., 2014*).
*Appendix 1—table 1* summarizes the full list of signature genes per cell type.

## Prediction of cell proportions in bulk samples with other tools

We compared EPIC's predictions with those from various cell fraction prediction methods. These other methods were run with the following packages (using the default options when possible):

- CIBERSORT (*Newman et al., 2015*) (R package version 1.03) was run based on two different gene expression reference profiles:
  - based on the *LM22* reference profiles derived in (*Newman et al., 2015*). For comparison with the experimentally measured cell proportions, we summed together the sub-types predictions of CIBERSORT within each major immune cell type.
  - based on the reference profiles and signature genes we derived here for EPIC.
- DeconRNASeq (v1.16) (*Gong and Szustakowski, 2013*) does not contain immune cell reference profiles and we used the reference profiles we derived here as well as the corresponding signature genes. We present the results with 'use.scale' parameter set to FALSE, which usually gave better results.
- DSA (*Zhong et al., 2013*) only needs a gene signature per cell type to estimate the proportion of cells in multiple bulk samples together. We used the implementation of DSA found in Cell-Mix (*Gaujoux and Seoighe, 2013*) R package (version 1.6.2). As DSA needs many samples to estimate simultaneously the proportions of cells in these samples, we considered all the PBMC samples from Hoek et al. data when fitting this dataset (eight samples) and all whole blood samples from Linsley et al. data when fitting this other dataset (20 samples), even though the cell proportions have been measured experimentally only for 2 and 5 samples respectively. For the gene signature, we used the same genes as those used for EPIC (*Appendix 1—table 1*; markers of B cells, CD4 T cells, CD8 T cells, monocytes, neutrophils and NK cells for the

predictions in the blood datasets; markers of B cells, CAFs, CD4 T cells, CD8 T cells, endothelial cells, macrophages, and NK cells for the predictions in solid tumors).

- ISOpure (*Quon et al., 2013*) estimates the profile and proportion of cancer cells by comparing many bulk samples containing cancer cells and many healthy bulk samples of the same tissue. Although the primary goal is not to compute the proportions of the different cell types composing a sample, cell fractions can still be obtained with this method. In particular, one output of ISOpure is how much each of the healthy reference samples is contributing to a given bulk sample. Instead of using bulk healthy samples, we used our cell reference profiles, so that each 'reference sample' corresponded to a different cell type. These reference samples and the bulk samples were then subsetted by the same signature genes that we derived for EPIC. ISOpure was then run based on these data. The contribution of each cell type was taken as the relative contribution outputted by ISOpure from each of the reference cell sample. The R implementation ISOpureR (*Anghel et al., 2015*) version 1.0.21 was used.
- MCP-counter (*Becht et al., 2016*) (R package version 1.1.0) was run with the '*HUGO_symbols*' chosen as features or with 'affy133P2_probesets' for the microarray-based IHC dataset.
- TIMER (*Li et al., 2016*) predictions were obtained by slightly adapting the available source code. The reference profiles available in TIMER were used directly. In addition to bulk gene expression, tumor purity estimates based on DNA copy number variation are needed in TIMER to refine the gene signature. As this information is not available in our benchmarking datasets, we kept all the original immune gene signatures for the predictions in blood. For the tumor datasets, we used the gene signatures obtained from the TCGA data for melanoma or colorectal cancer depending on the origin of cancer.
- ESTIMATE (*Yoshihara et al., 2013*) was run with their R package version 1.0.11.

For CIBERSORT, DeconRNASeq and ISOpure, when run based on our gene expression reference profiles, we used the reference profiles from peripheral blood immune cells for the predictions in blood and the reference profiles from tumor-infiltrating cells for the predictions in solid tumors.

## List of abbreviations

CAFs: cancer-associated fibroblasts; EPIC: acronym for our method to 'Estimate the Proportion of Immune and Cancer cells'; GEO: Gene Expression Omnibus; IHC: immunohistochemistry; PCA: principal component analysis; RMSE: root mean squared error; TCGA: The Cancer Genome Atlas; TPM: transcripts per million.

## Acknowledgements

We are grateful to Hélène Maby-El Hajjami for compiling the clinical data. We thank Kristen L. Hoek, Andrew Link and their colleagues (*Hoek et al., 2015*), Cate Speak, Scott Presnell and their colleagues (*Linsley et al., 2014*), Aurélien De Reynies, Etienne Becht and their colleagues (*Becht et al., 2016*), for providing us with additional data relating to their published studies. Computations were performed at the Vital-IT (http://www.vital-it.ch) Center for high-performance computing of the Swiss Institute of Bioinformatics.

## Additional information

### Funding

| Funder | Grant reference number | Author |
|---|---|---|
| Center for Advanced Modelling Science | | Julien Racle<br>David Gfeller |
| Schweizerischer Nationalfonds zur Förderung der Wissenschaftlichen Forschung | Project grant 31003A_173156 | Julien Racle<br>David Gfeller |

The funders had no role in study design, data collection and interpretation, or the decision to submit the work for publication.

## Author contributions
Julien Racle, Conceptualization, Data curation, Software, Formal analysis, Validation, Investigation, Visualization, Methodology, Writing—original draft, Writing—review and editing; Kaat de Jonge, Investigation, Methodology, Writing—review and editing; Petra Baumgaertner, Resources, Supervision, Investigation, Methodology; Daniel E Speiser, Resources, Supervision, Methodology, Writing—review and editing; David Gfeller, Conceptualization, Resources, Supervision, Funding acquisition, Validation, Methodology, Writing—original draft, Project administration, Writing—review and editing

## Author ORCIDs
Julien Racle, http://orcid.org/0000-0002-0100-0323
David Gfeller, http://orcid.org/0000-0002-3952-0930

## Ethics
Human subjects: Patients involved in this study agreed to donate metastatic tissues upon informed consent, based on dedicated clinical investigation protocols established according to the relevant regulatory standards. The protocols were approved by the local IRB, i.e. the Commission cantonale d'éthique de la recherche sur l'être humain du Canton de Vaud.

## Decision letter and Author response
Decision letter https://doi.org/10.7554/eLife.26476.047
Author response https://doi.org/10.7554/eLife.26476.048

# Additional files

## Supplementary files
• Supplementary file 1. Gene expression reference profiles, built from TPM (transcripts per million) normalized RNA-Seq data of immune cells sorted from blood as described in the Materials and methods: '*Reference gene expression profiles from circulating cells*'. The file includes two sheets: (A) the reference gene expression values; (B) the gene variability relating to the reference profile. Columns indicate the reference cell types; rows indicate the gene names.
DOI: https://doi.org/10.7554/eLife.26476.022

• Supplementary file 2. Gene expression reference profiles built from tumor-infiltrating cells obtained from TPM normalized single-cell RNA-Seq data as described in the Materials and methods: '*Reference profiles from tumor-infiltrating cells*'. The file includes two sheets: (A) the reference gene expression values; (B) the gene variability relating to the reference profile. Columns indicate the reference cell types; rows indicate the gene names.
DOI: https://doi.org/10.7554/eLife.26476.023

• Supplementary file 3. Proportion of cells measured in the different datasets: (A) this study; (B) dataset 1 (*Zimmermann et al., 2016*); (C) dataset 2 (*Hoek et al., 2015*); (D) dataset 3 (*Linsley et al., 2014*); and (E) single-cell RNA-Seq dataset (*Tirosh et al., 2016*). The 'Other cells' type corresponds always to the rest of the cells that were not assigned to one of the given cell types from the tables.
DOI: https://doi.org/10.7554/eLife.26476.024

• Transparent reporting form
DOI: https://doi.org/10.7554/eLife.26476.025

## Major datasets
The following dataset was generated:

| Author(s) | Year | Dataset title | Dataset URL | Database, license, and accessibility information |
|---|---|---|---|---|
| Racle J, de Jonge K, Baumgaertner P, Speiser DE, Gfeller D | 2017 | Simultaneous enumeration of cancer and immune cell types from tumor gene expression data | https://www.ncbi.nlm.nih.gov/geo/query/acc.cgi?acc=GSE93722 | Publicly available at the NCBI Gene Expression Omnibus (accession no: GSE93722) |

The following previously published datasets were used:

| Author(s) | Year | Dataset title | Dataset URL | Database, license, and accessibility information |
|---|---|---|---|---|
| Poland G | 2015 | Bioinformatics Approach to 2010-2011 TIV Influenza A/H1N1 Vaccine Immune Profiling | http://www.immport.org/immport-open/public/study/study/displayStudyDetail/SDY67 | Available at ImmPort (accession no: SDY67) |
| Hoek KL, Link AJ | 2015 | A Cell-based Systems Biology Assessment of Human Blood to Monitor Immune Responses After Influenza Vaccination | https://www.ncbi.nlm.nih.gov/geo/query/acc.cgi?acc=GSE64655 | Publicly available at the NCBI Gene Expression Omnibus (accession no: GSE64655) |
| Speake C, Linsley PS, Whalen E, Chaussabel D, Presnell SR, Mason MJ, Gersuk VH, O'Brien KK, Nguyen Q, Greenbaum CJ, Buckner JH, Malhotra U | 2015 | Next generation sequencing of human immune cell subsets across diseases | https://www.ncbi.nlm.nih.gov/geo/query/acc.cgi?acc=GSE60424 | Publicly available at the NCBI Gene Expression Omnibus (accession no: GSE60 424) |
| Sauvageau G, Pabst C, Yeh J | 2016 | RNA-Seq analysis of human adult peripheral blood populations | https://www.ncbi.nlm.nih.gov/geo/query/acc.cgi?acc=GSE51984 | Publicly available at the NCBI Gene Expression Omnibus (accession no: GSE51984) |
| Marisa L, de Reyniès A, Duval A, Selves J, Gaub M, Vescovo L, Etienne-Grimaldi M, Schiappa R, Guenot D, Ayadi M, Kirzin S, Chazal M, Fléjou J, Benchimol D, Pencreach E, Lagarde A, Piard F, Elias D, Olschwang S, Milano G, Laurent-Puig P, Boige V | 2013 | Gene expression Classification of Colon Cancer defines six molecular subtypes with distinct clinical, molecular and survival characteristics [Expression] | https://www.ncbi.nlm.nih.gov/geo/query/acc.cgi?acc=GSE39582 | Publicly available at the NCBI Gene Expression Omnibus (accession no: GSE39582) |
| Tirosh I, Izar B | 2016 | Single cell RNA-seq analysis of melanoma | https://www.ncbi.nlm.nih.gov/geo/query/acc.cgi?acc=GSE72056 | Publicly available at the NCBI Gene Expression Omnibus (accession no: GSE72056) |

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

# Appendix 1

DOI: https://doi.org/10.7554/eLife.26476.026

**Appendix 1—table 1. Gene markers used per cell type.** Only markers of cell types present in the respective reference gene expression profiles are used.

| Cell type | Genes markers |
|---|---|
| B cells | BANK1, CD79A, CD79B, FCER2, FCRL2, FCRL5, MS4A1, PAX5, POU2AF1, STAP1, TCL1A |
| CAFs | ADAM33, CLDN11, COL1A1, COL3A1, COL14A1, CRISPLD2, CXCL14, DPT, F3, FBLN1, ISLR, LUM, MEG3, MFAP5, PRELP, PTGIS, SFRP2, SFRP4, SYNPO2, TMEM119 |
| CD4 T cells | ANKRD55, DGKA, FOXP3, GCNT4, IL2RA, MDS2, RCAN3, TBC1D4, TRAT1 |
| CD8 T cells | CD8B, HAUS3, JAKMIP1, NAA16, TSPYL1 |
| Endothelial cells | CDH5, CLDN5, CLEC14A, CXorf36, ECSCR, F2RL3, FLT1, FLT4, GPR4, GPR182, KDR, MMRN1, MMRN2, MYCT1, PTPRB, RHOJ, SLCO2A1, SOX18, STAB2, VWF |
| Macrophages | APOC1, C1QC, CD14, CD163, CD300C, CD300E, CSF1R, F13A1, FPR3, HAMP, IL1B, LILRB4, MS4A6A, MSR1, SIGLEC1, VSIG4 |
| Monocytes | CD33, CD300C, CD300E, CECR1, CLEC6A, CPVL, EGR2, EREG, MS4A6A, NAGA, SLC37A2 |
| Neutrophils | CEACAM3, CNTNAP3, CXCR1, CYP4F3, FFAR2, HIST1H2BC, HIST1H3D, KY, MMP25, PGLYRP1, SLC12A1, TAS2R40 |
| NK cells | CD160, CLIC3, FGFBP2, GNLY, GNPTAB, KLRF1, NCR1, NMUR1, S1PR5, SH2D1B |
| T cells | BCL11B, CD5, CD28, IL7R, ITK, THEMIS, UBASH3A |

DOI: https://doi.org/10.7554/eLife.26476.027

# Appendix 2

DOI: https://doi.org/10.7554/eLife.26476.028

**Appendix 2—table 1. Characteristics of the patients with metastatic melanoma and corresponding lymph node samples.**

| Patient | Age (years) | Gender | Tissue |
| --- | --- | --- | --- |
| LAU125 | 59 | male | iliac lymph node |
| LAU355 | 70 | female | iliac-obturator lymph node |
| LAU1255 | 87 | male | axillary lymph node |
| LAU1314 | 81 | male | iliac-obturator lymph node |

DOI: https://doi.org/10.7554/eLife.26476.029

