## [Decision Letter]

Thank you for submitting your article "Simultaneous enumeration of cancer and immune cell types from bulk tumor gene expression data" for consideration by *eLife*. Your article has been favorably evaluated by Naama Barkai (Senior Editor) and three reviewers, one of whom is a member of our Board of Reviewing Editors. The reviewers have opted to remain anonymous.

The reviewers have discussed the reviews with one another and the Reviewing Editor has drafted these comments to understand if it will be possible for you to provide what the reviewers consider essential to the publication of this work. At this point we ask you to provide a detailed response with an action plan and timetable for the completion of the essential tasks noted below. We will have the Board and reviewer consider your responses and issue a binding recommendation as soon as possible.

Summary:

This manuscript describes the EPIC method to estimate the proportion of cancerous cells and immune cells of various types within a bulk gene expression profile of a tumor sample or PBMC. The work fits within the recent renewed interest in the area of expression deconvolution in tumor-infiltrating lymphocytes.

The method is based on the extrapolation of the proportion of RNA in standard cell types to determine the proportion of cell types in mixtures. The main features of the method are: it takes into account the estimated proportion of RNA in the samples, weights highly the contribution of less variable genes in the reference set, it does not make assumptions about the heterogeneity of the non-normal cells in the samples, and, it uses the expression of maker genes that are not expressed in cancer cells to normalize the proportion of tumor/non tumor cells in the samples. The data presented correspond mainly to metastatic melanoma samples with some additional colon cancer cases.

Essential revisions:

1) The works requires a clear demonstration of the capacity of the method to do well estimating cell subsets at higher resolution, as this seems to be the main advantage of the method.

2) A better analysis and clarification of what are the key steps of the procedure that improve the results.

3) Clarify the dependency of the results of outliers and in general revise the significance criteria.

4) Make the system accessible via a web interface.

Other concerns:

1) – The authors should consider to extend the reference gene expression profiles to additional cell types such as stromal and endothelial cells which are known to have a role in tumor microenvironment as well as deepen the separation between cytotoxic T cells (CD8 cells) and CD4 T cells gene expression profiles, which currently is low and at a level less than what has been shown to be important correlates for immune-tumor interactions/therapeutic response.

2) Consider rephrasing, the estimation of the tumor in 3D – it is not too surprising given that the melanoma forms the bulk of the tumor in most of these samples and is quite distinct – this is akin to getting neutrophil estimates correct in whole blood.

3) With respect to the second validation (comparing EPIC to IHC data), the authors claimed they observed "a good agreement between cell proportions measured by IHC and our predictions", however there is a weak non-significant correlation (r=0.3, p=0.09) between these factors in macrophages that should be mentioned in the text. This may be due to the maker used in IHC being expressed in other cell subsets or due to protein/mRNA differences – the former being relatively easy to distinguish.

4) The first test and comparison with other methods is carried out with normal blood samples and (Figure 2). The EPIC method performs better (pearson R) than other methods. Notice that in whole blood (dataset 3 (Linsley et al., 2014)) the differences with CIBERSORT and DSA are minimal. It will be important to clarify if the additional information is provided by the inclusion of the estimated mRNA content in the calculations (Figure 2 versus 2C).

5) Reconsider the following claim "We observed a remarkable agreement between our predictions and experimentally determined cell fractions (Figure 3). Of note, the proportion of melanoma cells could be very accurately predicted even in the absence of a priori information about their gene expression." in light of the small size of the data sets (4 donors of which two of them dominate the correlation).

6) Comment of the significance of the results in Figure 3 and Figure 3 where the differences are based on single outliers.

7) It will be good to check if in the information in Figure 5 (and supplementary figures) can be better interpreted comparing the method only in normal tissue samples (equivalent to Figure 5 supplementary figures).

8) Assess the significance of the results in Figure 5.

9) Benchmarking other methods, such as Cibersort by summing high-resolution cell subsets may be incorrect – a better approach would be to replace LM22 with the matrix derived by the authors – if not, I would suggest saying that this comparison is less than ideal.

10) It is not clear from the manuscript whether or not the advance over CIBERSORT is due to a better marker gene set and reference panel or due to the other methodological innovations that they made. This point should be clarified.

11) The reference to the use of "zeros" in the gene expression profile of cancerous tissue as informative to be leveraged in deconvolution in Gosink 2007 and Clarke 2010 should be included.

12) Respect to ISOpure, it is not clear from the description whether ISOpure was run just with signature genes or with the whole gene expression profile.

13) The ideas implicit in DeMix where the ratio of the variances provides an implicit importance weight (Ahn 2013) should be discussed.

Other points:

1) The authors should change the scale and marking of the statistical significance from (*p<0.1; **p<0.05; ***p<0.01) to (*p<0.05; **p<0.01; ***p<0.001) as p>0.05 is not considered significant.

2) The authors claimed that "Immune cells differ in their gene expression profiles depending on their state and site of origin (e.g., blood or tumors)" – a reference should be added.

3) There is a lack in specifying "Supplementary file 4" in the relevant places.

4) Figures and supplementary figures are not numbered in the manuscript, what makes difficult to know what the supplementary figures correspond to what text.

5) The use of a different scale for each method does not help to the interpretation of the figures.

6) Subsection “Validation in blood”: Renormalization by mRNA content is used in (Qiao 2012) to improve the estimates of cell-type proportions.

7) "Prediction of cancer and immune cell type proportions": I found this explanation somewhat confusing because Equation 1 is never actually used in the deconvolution – only Equations 6 and 7. Because Equation 1 is introduced, I expected EPIC to try to solve it at some point.

Regarding references:

1) The authors claim that this is the first deconvolution methodology to take into account total RNA differences between cell subsets is incorrect, to the best of this reviewers knowledge, Baron et al. (Cell Systems 2016) propose a method, Bseq-SC which uses total RNA to improve deconvolution estimates. Of relevance here, their estimates for total RNA levels come from the single cell expression data, information which could be incorporated into EPIC too to increase its resolution.

Other pertinent references include:

Clarke J, Seo P, Clarke B: Statistical expression deconvolution from mixed tissue samples. Bioinformatics. 2010, 26: 1043-1049

Gosink MM, Petrie HT, Tsinoremas NF: Electronically subtracting expression patterns from a mixed cell population. Bioinformatics. 2007, 23: 3328-3334.

Ahn J, Yuan Y, Parmigiani G, Suraokar MB, Diao L, Wistuba II, Wang W. DeMix: deconvolution for mixed cancer transcriptomes using raw measured data. Bioinformatics. 2013 Aug 1;29(15):1865-71.

Qiao W, Quon G, Csaszar E, Yu M, Morris Q, Zandstra PW. PERT: a method for expression deconvolution of human blood samples from varied microenvironmental and developmental conditions. PLoS Comput Biol. 2012

[Editors' note: further revisions were requested prior to acceptance, as described below.]

Thank you for resubmitting your work entitled "Simultaneous enumeration of cancer and immune cell types from bulk tumor gene expression data" for further consideration at *eLife*. Your revised article has been favorably evaluated by Naama Barkai (Senior Editor) and three reviewers, one of whom is a member of our Board of Reviewing Editors.

The manuscript has been improved but there are some remaining issues that need to be addressed before acceptance, as outlined below:

Essential revisions:

The claim that "ISOpure is specifically designed to run genome-wide" does not appear anywhere in the ISOpure manuscript, is not consistent with the uses of ISOpure, and there is nothing obvious in the ISOpure algorithm that makes it specifically designed to be applied genome-wide (such a claim is made for PERT, and one could argue that PERT contains a gene-specific weighting mechanism, like EPIC, which makes it suitable for genome-wide use, but no such claim is made for ISOpure).

Indeed, an ISOpure-like method applied genome-wide in the PERT manuscript does quite badly at recovering the mixture proportions of the various cell types. Therefore, the new version in the revised version regarding ISOpure needs to be revised and inaccurate statements removed.

In conclusion, in the new version EPIC should be compared with ISOpure using the same inputs as CIBERSORT and EPIC.

---

## [Author Response]

Essential revisions:1) The works requires a clear demonstration of the capacity of the method to do well estimating cell subsets at higher resolution, as this seems to be the main advantage of the method.

We agree that it is important to estimate cell subsets at a higher resolution. For this reason, we have now constructed gene expression reference profiles for CD4 T cells and CD8 T cells from the peripheral blood datasets. In addition, we also took advantage of the single-cell RNA-seq data from Tirosh et al., 2016, in order to define these CD4 and CD8 T cells for tumor infiltrating cells. Finally, we have built reference gene expression profiles from stromal cells (cancer associated fibroblasts) and endothelial cells, also from the single-cell RNA-Seq data. As before, we defined signature genes for these new cell types.

Going back to the FACS data from the study of Zimmermann et al., 2016, we could estimate the proportion of CD4 T cells and CD8 T cells that were present in these blood samples, and used this data as a first validation of EPIC in blood data, including these new cell subsets, as shown in Author response image 1.

**Author response image 1. respfig1:** Comparison between EPIC predictions and measured cell fractions in PBMC dataset from Zimmermann et al. 2016.

Concerning the validation in tumor samples:

1) We expanded our flow cytometry analysis of the four melanoma samples, including CD4 T cells and CD8 T cells to test the predictions there.

2) We could also validate EPIC predictions for the CD8 T cells in the immunohistochemistry data from Becht et al., 2016.

3) Predictions from all cell types, including stromal and endothelial cells, could also be validated based on the data from Tirosh et al., 2016.

We performed a detailed comparison of the predictions from EPIC and the other methods, including these additional cell types. Some of the results are given in Author response image 2, showing the correlation between the predictions from EPIC and the experimentally observed proportions for the various cell types in different datasets.

**Author response image 2. respfig2:** Comparison between the experimentally measured cell fractions and EPIC predictions, including additional cell types in: (**A**) our expanded flow cytometry analysis of melanoma; (**B**) lymph node metastasis and primary tumor melanoma data from Tirosh et al., 2016.

Pushing the limits of the cell type fraction predictions to even lower subsets, we also defined Thelper and Treg cell subsets, based on the data from Tirosh et al., 2016. We note that in such cases, the accuracy of the predictions starts to suffer because of the substantial similarity of these cell types at the transcriptional level. The results based on EPIC are nevertheless still as good or even better than those obtained from CIBERSORT (Figure 5—figure supplement 6). Our results suggest that quantitative predictions for such less abundant and more redundant cell types are still challenging and may reach the limits of the signal to noise ratio in gene expression data.

We still note that, in our view, estimating cell subsets at higher resolution, albeit very promising, may not be the main advantage of EPIC compared to other methods. Rather, we believe that inclusion of more relevant reference profiles (tumor infiltrating vs. blood), the ability to consider uncharacterized cell types and the renormalization by mRNA content are more important for the improvement in predictions.

2) A better analysis and clarification of what are the key steps of the procedure that improve the results.

We are sorry we did not satisfactorily detail the improvements brought by each of the steps from EPIC.

We have therefore complemented our analyses, clarifying the key steps of the procedure improving the results. The two most important steps are:

1) Renormalization by mRNA content (Figure 2 vs. Figure 2; Figure 2—figure supplement 2. Figure 2—figure supplement 3).

2) Ability to consider uncharacterized (and possibly highly variable) cell types in the cell fraction prediction (Figure 2—figure supplement 4; and the comparisons in Figure 5—figure supplement 1–Figure 5—figure supplement 5).

Other steps, like the use of the variability in the gene expression reference profiles (Figure 2—figure supplement 3) and the use of more biologically relevant reference gene expression profiles (Figure 3 vs. Figure 4, Figure 4—figure supplement 1) further add some improvements to the results, as discussed in the manuscript.

3) Clarify the dependency of the results of outliers and in general revise the significance criteria.

We agree that outliers could have some influence on the results. For this reason, we decided to remove the analysis performed about the melanoma immunohistochemistry data from Jönsson et al., 2010. Indeed this dataset was only qualitative, grouping the B and T cells infiltration into absent, low or high, and within the high groups, only four samples were present, two of which were very highly infiltrated such that results were highly biased by these. If removing these two outlier samples, then not enough samples remained in the high infiltration to be conclusive.

Considering the other results, we revised our significance criteria, lowering the p-values needed for results to be considered as significant.

We also discussed in the updated manuscript that datasets with a large range of data points (i.e. large variability in cell fractions across samples) could present some very high correlations due to few points driving the correlation. We then emphasized that it is important to consider the correlation values only to compare the different methods on the same dataset as the range of values there will be the same.

In addition to the correlation values, we also underline in the revised manuscript that the root mean squared values provide another good indicator of the quality of the predictions. Notably, the RMSE is not influenced by outlier data points that can drive the high correlation values.

Based on all these comparisons, we still believe that EPIC performed very well and often significantly better than other methods.

4) Make the system accessible via a web interface.

We have now implemented such a web interface and made it available at the address: http://epic.gfellerlab.org.

Users can upload their own dataset for which they want to estimate the proportion of cells. They could then choose to use our implemented gene expression reference profiles or to upload some reference profiles of their own, and define various other parameters.

Results are then outputted as a web-table with the various cell type proportions (this table can also easily be downloaded), and different figures complement the results, giving an overview of the various cell type proportions.

Other concerns:1) The authors should consider to extend the reference gene expression profiles to additional cell types such as stromal and endothelial cells which are known to have a role in tumor microenvironment as well as deepen the separation between cytotoxic T cells (CD8 cells) and CD4 T cells gene expression profiles, which currently is low and at a level less than what has been shown to be important correlates for immune-tumor interactions/therapeutic response.

We have now extended our reference gene expression profiles to also include also CD4^+^ T cells, CD8^+^ T cells, stromal (cancer-associated fibroblasts) and endothelial cells, showing good accuracy of the predictions for these cell types as well, as described above in point 1.

2) Consider rephrasing, the estimation of the tumor in 3D – it is not too surprising given that the melanoma forms the bulk of the tumor in most of these samples and is quite distinct – this is akin to getting neutrophil estimates correct in whole blood.

We have updated our discussion of these results in the subsection “Validation in blood” to clarify why these predictions can really be considered as good and are not just an artifact of the cancer cell gene expression being totally different from the other cell types.

We emphasize here that it is not only the correlation that is good in our predictions but also the root mean squared error, showing that our predictions are able to predict accurately the true absolute proportions of the melanoma cells (e.g. in Figure 3 (previous 3D), the predicted proportion of cells fall nearly perfectly on the gray dashed line, which corresponds to the y = x line (i.e. line of "predicted value" = "observed value").

In addition to having good predictions for the melanoma cell fractions, these results also show that the predictions were also accurate for the various immune cell types, which have much more similar reference profiles (Figure 4—figure supplement 1), and thus the results we present there are not just as simple as if we would only be predicting the neutrophils proportions from a whole blood sample.

3) With respect to the second validation (comparing EPIC to IHC data), the authors claimed they observed "a good agreement between cell proportions measured by IHC and our predictions", however there is a weak non-significant correlation (r=0.3, p=0.09) between these factors in macrophages that should be mentioned in the text. This may be due to the maker used in IHC being expressed in other cell subsets or due to protein/mRNA differences – the former being relatively easy to distinguish.

In our revised predictions, the correlation between the macrophages predicted by EPIC and observed IHC values is better, even with the reference profiles from peripheral blood (new correlation is r=0.52, p=0.002). This improvement can be explained by our use of a refined list of signature genes and the use of TPM-normalized reference profiles.

However the results are not significant in this analysis for the predictions of the CD8 T cells that did not exist in the previous version of the manuscript. We made therefore a comment in the revision about this correlation that is not significant.

Importantly however, the correlations for both macrophages and CD8 T cells improve in this immunohistochemistry dataset when using the reference profiles from tumor infiltrating cells (r=0.66, p<10^-4^ and r=0.53, p=0.001 respectively). We anticipate that this improvement is due to the use of reference gene expression profiles corresponding better to the phenotype of CD8 T cells and macrophages present in primary tumors.

We expanded the discussion about this point in the Discussion section.

4) The first test and comparison with other methods is carried out with normal blood samples and (Figure 2). The EPIC method performs better (pearson R) than other methods. Notice that in whole blood (dataset 3 (Linsley et al., 2014)) the differences with CIBERSORT and DSA are minimal. It will be important to clarify if the additional information is provided by the inclusion of the estimated mRNA content in the calculations (Figure 2 versus 2C).

Our new comparisons show that indeed other methods like CIBERSORT and DSA could also benefit from the renormalization by mRNA (Figure 2—figure supplement 2).

Importantly however, EPIC is especially designed to predict the proportion of cells in tumor samples and the ability of EPIC to consider a cell type without any reference gene expression profiles is another particularly important improvement in this context (Figure 2—figure supplement 4).

5) Reconsider the following claim "We observed a remarkable agreement between our predictions and experimentally determined cell fractions (Figure 3). Of note, the proportion of melanoma cells could be very accurately predicted even in the absence of a priori information about their gene expression." in light of the small size of the data sets (4 donors of which two of them dominate the correlation).

We have complemented the discussion of these results, underlining this possible artifact. We still point out that, despite this small sample size and some data points that might be driving high correlation values, the predictions fall nearly on the y=x line, indicating that all cell types could indeed be predicted accurately, as also reflected by the low RMSE.

In addition, we emphasized in the revised manuscript that the correlation values should only be compared for methods tested on the same datasets, and in our comparison with the other methods these correlations were lower for the other methods, despite the presence of the same data possibly dominated by two donors.

6) Comment of the significance of the results in Figure 3 and Figure 3 where the differences are based on single outliers.

We agree that outliers in Figure 3 have some influence on the results. For this reason, as detailed in the response to major comment 3), we decided to remove the analysis performed about this data. But we do not think that outliers in Figure 3 are biasing the results from this other analysis.

7) It will be good to check if in the information in Figure 5 (and supplementary figures) can be better interpreted comparing the method only in normal tissue samples (equivalent to Figure 5 supplementary figures).

We believe this concern has already been answered. Indeed in Figure 5 we do a comparison of all the methods if considering only the normal immune cell types. And in Figure 2 (and Figure 2—figure supplement 1–Figure 2—figure supplement 2) we performed a detailed comparison of all methods based on healthy blood samples.

8) Assess the significance of the results in Figure 5.

We have now indicated with some stars the p-values for each correlation given on Figure 5 as well as on Figure 5 and Figure 2.

9) Benchmarking other methods, such as Cibersort by summing high-resolution cell subsets may be incorrect – a better approach would be to replace LM22 with the matrix derived by the authors – if not, I would suggest saying that this comparison is less than ideal.10) It is not clear from the manuscript whether or not the advance over CIBERSORT is due to a better marker gene set and reference panel or due to the other methodological innovations that they made. This point should be clarified.

These two comments are highly related in will thus be answered together.

First, we have now included in our analyses the results from CIBERSORT based on LM22 reference profiles as well as based on the reference profiles and signatures we derived in our study. We can note that the results from CIBERSORT based on either of the reference profiles are similar; in some cases results are better based on LM22 signature and in other cases it is better with our signature (e.g. Figure 2, Figure 2—figure supplement 1, Figure 5).

We also showed in the new Figure 2—figure supplement 1-4 that the methodological innovations we implemented in EPIC are important for accurate predictions, which is then verified in the context of tumor samples as exemplified by the results from Figure 5 and Figure 5—figure supplement 1.

11) The reference to the use of "zeros" in the gene expression profile of cancerous tissue as informative to be leveraged in deconvolution in Gosink 2007 and Clarke 2010 should be included.

We thank the reviewer for pointing out these papers and we have included references to these in the revised manuscript, discussing their idea of using genes not expressed by a given cell to estimate its proportion.

12) Respect to ISOpure, it is not clear from the description whether ISOpure was run just with signature genes or with the whole gene expression profile.

We apologize if the explanation on how ISOpure was run was not clear. Indeed, we used here the whole gene expression profiles and not only the signature genes, as ISOpure has been specifically designed to account for the gene expression of all the genes from a sample. This point has been clarified in the text.

13) The ideas implicit in DeMix where the ratio of the variances provides an implicit importance weight (Ahn 2013) should be discussed.

We thank the reviewer for this reference and we added it in the text. This is also some interesting framework to estimate the proportion of cell types from a mixed sample in the presence of one uncharacterized cell type. This framework works in a similar way than ISOpure that we have tested in our manuscript.

The weights given to each gene in EPIC (based on their variability in the reference profiles) is included in the constrained least square optimization we perform. In contrary, DeMix method does not perform such regression based on signature genes, but it considers the distribution of each gene in each pure normal tissue sample and in each tumor sample. Based on these distributions, assumed to be log-normal distributed in the normal tissue, DeMix estimates the gene distribution inside the tumor cells and proportion of tumor cells that would best match the observed distribution of genes in the mixed tumor samples. We agree that when doing such predictions, the genes that had a narrower distribution both in the normal and tumor samples will be implicitly more important and should have a bigger impact on the predicted tumor cell fractions. However, no real weights will be given to these genes in solving the problem with DeMix.

Also, this method assumes that the normal tissue is composed of a single type of “normal” cells or at least that the normal cell types have a constant proportion inside normal tissue. Such a framework would thus not be appropriate to study the proportions of various immune cell types infiltrating a tumor, as it has been observed that the proportion of the different normal cells (various immune cell types, stromal and endothelial cells) vary from one patient to another.

Other points:1) The authors should change the scale and marking of the statistical significance from (*p<0.1; **p<0.05; ***p<0.01) to (*p<0.05; **p<0.01; ***p<0.001) as p>0.05 is not considered significant.

As suggested by the reviewer, we modified the scale of statistical significance to (*p<0.05; **p<0.01; ***p<0.001) on the figures where the exact p-value was not directly given.

2) The authors claimed that "Immune cells differ in their gene expression profiles depending on their state and site of origin (e.g., blood or tumors)" – a reference should be added.

We added various references that discuss the fact that tumor infiltrating immune cells present an altered phenotype compared to circulating immune cells (e.g. describing the exhaustion phenotype of T cells).

3) There is a lack in specifying "Supplementary file 4" in the relevant places.

We have added references to this file (now named Supplementary file 3) in the legend of the respective figures. References to this file were already present in various places in the main text.

4) Figures and supplementary figures are not numbered in the manuscript, what makes difficult to know what the supplementary figures correspond to what text.

We are sorry that the numbers did not appear in the figure. This has now been corrected and all figures are numbered.

5) The use of a different scale for each method does not help to the interpretation of the figures.

The scale of the different figures are now the same for each method except for MCPcounter. Indeed MCPcounter outputs do not correspond to cell fractions so that its exact values are not comparable to the other methods. Then, the scales are not necessarily the same between various datasets due to different ranges in the data present in each dataset. Also for the immunohistochemistry data from Becht et al., 2016 (Figure 5—figure supplement 3), the ranges in the predicted proportions from the various methods is broad and we thus use different scales for each method to better see the results, as the true cell proportions are not known (only the intensity of some cell-specific protein markers are known from the immunohistochemistry).

6) Subsection “Validation in blood”: Renormalization by mRNA content is used in (Qiao 2012) to improve the estimates of cell-type proportions.

We included a reference to the method presented in this paper, PERT, which is similar to the method ISOpure we have tested here. To our understanding, ISOpure is some further improvement over PERT, developed in the same group, in order to allow for the estimation of an uncharacterized cell type in addition to the known cell types.

Also, to our understanding, PERT is allowing the gene expression value from each gene to be multiplied by some factor (different for each gene), in order to account for differences in expression in the reference cells studied in isolation and the same cells present in a mixed sample. However, it seems to us that PERT is not considering any renormalization by mRNA content at the cell level.

7) "Prediction of cancer and immune cell type proportions": I found this explanation somewhat confusing because Equation 1 is never actually used in the deconvolution – only Equations 6 and 7. Because Equation 1 is introduced, I expected EPIC to try to solve it at some point.

We are sorry if the description of the equation was not very clear, but we think that this equation (1) is actually very important, as this is the equation this is at the root of EPIC method (see e.g. Figure 1). In particular, equations (6) and (7) are directly resulting from equation (1). These two equations are obtained from equation (1) after normalizing the data into TPM as described in equation (2) and restraining the resolution to a subset of signature genes that are not expressed by the cancer cells, which simplify the system to solve. In order to make the explanation more easily understandable, we have decided to use the standard TPM normalization instead of our previous normalization of the counts. We have also extended equation (6) to detail how it is finally obtained.

Regarding references:1) The authors claim that this is the first deconvolution methodology to take into account total RNA differences between cell subsets is incorrect, to the best of this reviewers knowledge, Baron et al. (Cell Systems 2016) propose a method, Bseq-SC which uses total RNA to improve deconvolution estimates. Of relevance here, their estimates for total RNA levels come from the single cell expression data, information which could be incorporated into EPIC too to increase its resolution.

We thank the reviewer for pointing out this reference and apologize for not citing this paper, which we had overlooked.

In this paper, the authors study normal pancreas at the single-cell level, having sequenced a very large amount of human and mice cells. In their analysis, they also developed a method, BSeq-SC to estimate the proportion of the various type of pancreatic cells from a bulk sample. This method integrates reference profiles the authors built from their single-cell RNA-seq data and then use CIBERSORT for the cell fraction predictions. The authors indeed observed that the different types of cells (mostly pancreatic but also very few immune cells) contained a different number of total transcripts. They used this information when building the reference profiles from pancreatic cells.

Contrary to EPIC, this renormalization by mRNA content is thus performed a priori on the reference gene expression profiles, instead of doing it a posteriori on the predicted mRNA proportions. For completeness in our analyses, we thus also modified EPIC in order to perform such a priori renormalization and observed the results were similar to those based on the a posteriorinormalization developed for EPIC (Figure 2—figure supplement 3).

[Editors' note: further revisions were requested prior to acceptance, as described below.]

Essential revisions:The claim that "ISOpure is specifically designed to run genome-wide" does not appear anywhere in the ISOpure manuscript, is not consistent with the uses of ISOpure, and there is nothing obvious in the ISOpure algorithm that makes it specifically designed to be applied genome-wide (such a claim is made for PERT, and one could argue that PERT contains a gene-specific weighting mechanism, like EPIC, which makes it suitable for genome-wide use, but no such claim is made for ISOpure).Indeed, an ISOpure-like method applied genome-wide in the PERT manuscript does quite badly at recovering the mixture proportions of the various cell types. Therefore, the new version in the revised version regarding ISOpure needs to be revised and inaccurate statements removed.In conclusion, in the new version EPIC should be compared with ISOpure using the same inputs as CIBERSORT and EPIC.

We thank the reviewer for this insightful comment about how to best use ISOpure method and apologize for the confusion between PERT and ISOpure. We have now re-run this method based on the same subset of signature genes than what we had derived for EPIC. Of note, these signature genes (in addition to the reference profiles) are a new list of signature genes that we have derived in this manuscript for EPIC but they can be used in other contexts as well, like here for ISOpure.

We observe some improvement for ISOpure based on such subset of genes compared to previous whole-genome results (see e.g. the figures in Author response image 3 comparing the previous results and updated ones). For this reason, we included the revised gene-signature based results for ISOpure in our manuscript.

**Author response image 3. respfig3:** Comparison of the prediction accuracies for EPIC, ISOpure based on all genes and ISOpure based on the subset of signature genes we derived for EPIC. (**A**) For all immune cell types in the blood datasets (dataset 1: Zimmermann et al. 2016; dataset2: Hoek et al. 2015; dataset 3: Linsley et al. 2014). (**B**) and (**C**) in the tumor datasets, based on all cell types, including immune, stromal and cancer cells (**B**), or based only on all the immune cell types (**C**) (flow cytometry: our new experiment; single-cell RNA-seq: data from Tirosh et al. 2016). The stars above each bar indicate if the Pearson correlation was significant (* p < 0.05; ** p < 0.01; *** p < 0.001). These figures are the same than in our manuscript Figure 2 and Figure 5 but comparing different ISOpure results and EPIC ones.

We can note that despite this improvement for ISOpure, EPIC is still performing better. It is important to underline that the main goal of ISOpure is to estimate the proportion of the cancer cells present in the bulk samples and to estimate the gene expression profiles from the unknown cancer cells from these bulk samples, as we have also written in the manuscript. This can be a reason why ISOpure performs less good than EPIC to predict the proportion of all cell types present in the bulk as this is not the main focus of ISOpure contrary to EPIC.

Finally, please note that the reason why we had originally used ISOpure based on all genes is that the method description of ISOpure's manuscript (Quon et al. 2013) states that the tumor and healthy profiles are composed of the measured gene expression level of *G* transcripts, where "*G* is typically on the order of 10,000". It seemed to us that a whole genome data would better reflect these numbers than a subset of about hundred signature genes, and we had thought that ISOpure would therefore perform better based on all genes, which we have now seen was not really the case here.